# Inverse resource allocation between vision and olfaction across the genus *Drosophila*

Ian W. Keesey [1], Veit Grabe[1], Lydia Gruber[1], Sarah Koerte[1], George F. Obiero[1,2], Grant Bolton[3], Mohammed A. Khallaf[1], Grit Kunert[4], Sofia Lavista-Llanos[1], Dario Riccardo Valenzano[5], Jürgen Rybak[1], Bruce A. Barrett[3], Markus Knaden[1] & Bill S. Hansson[1]

Divergent populations across different environments are exposed to critical sensory information related to locating a host or mate, as well as avoiding predators and pathogens. These sensory signals generate evolutionary changes in neuroanatomy and behavior; however, few studies have investigated patterns of neural architecture that occur between sensory systems, or that occur within large groups of closely-related organisms. Here we examine 62 species within the genus *Drosophila* and describe an inverse resource allocation between vision and olfaction, which we consistently observe at the periphery, within the brain, as well as during larval development. This sensory variation was noted across the entire genus and appears to represent repeated, independent evolutionary events, where one sensory modality is consistently selected for at the expense of the other. Moreover, we provide evidence of a developmental genetic constraint through the sharing of a single larval structure, the eye-antennal imaginal disc. In addition, we examine the ecological implications of visual or olfactory bias, including the potential impact on host-navigation and courtship.

[1] Max Planck Institute for Chemical Ecology, Department of Evolutionary Neuroethology, Hans-Knöll-Straße 8, D-07745 Jena, Germany. [2] Department of Biochemistry and Biotechnology, Technical University of Kenya, Haille-Sellasie Avenue, Workshop Road, 0200 Nairobi, Kenya. [3] University of Missouri, Division of Plant Sciences, 3-22I Agriculture Building, Columbia, Missouri 65211, USA. [4] Max Planck Institute for Chemical Ecology, Department of Biochemistry, Hans-Knöll-Straße 8, D-07745 Jena, Germany. [5] Max Planck Institute for Biology of Ageing and CECAD at University of Cologne, Joseph-Stelzmann-Str 9b and 26, Cologne 50931, Germany. These authors jointly supervised this work: Markus Knaden, Bill S. Hansson. Correspondence and requests for materials should be addressed to M.K. (email: mknaden@ice.mpg.de) or to B.S.H. (email: hansson@ice.mpg.de)

A pivotal question in neuroscience focuses on how the morphology and structure of the brain relates to its function and thereby its behavioral relevance. Neuroscience in general utilizes a wide array of techniques, including both genetics and neuroanatomical imaging, in order to unravel neural mechanisms underlying animal behavior and to understand how these circuits translate into the natural behaviors that are associated with an animal's specific ecological niche, for example, in regard to decisions concerning host navigation or mate selection[1].

One of the ultimate goals of neuroethology is to understand the principles organizing and defining these complex neural circuits, both from an ecological as well as an evolutionary perspective, and to decipher how the brain processes information while guiding behavioral responses toward naturally occurring stimuli. Previous research has supported the notion that structural size in a sensory phenotype correlates with its functional significance, for example, the reduction of sight in cave fish[2,3], the enlarged ears of echolocating bats[4–6], or the enlarged eyes of predatory birds[7]. Moreover, neuroanatomical studies have also shown that the size of each brain region corresponds to the organism's morphological specialization, thus for example, the smaller the eyes, the less importance of visual stimuli, and the smaller the brain region dedicated toward vision[2,3]. Other studies have also sought to associate sensory size with behavioral or ecological importance, such as the enlarged male-specific macroglomerular complex (MGC) in the Lepidoptera[8,9], the enlarged DM2 glomerulus in *Drosophila sechellia*[10], or an enlarged glomerulus based on the number of OSNs or synapses[11,12]. In each of these cases, the enlarged structure is indicative of the importance of a particular ecological stimulus, and moreover, that the relative morphological size of a sensory structure relates to its importance. However, just as studying a single neuron will not be sufficient to understand the function of the whole brain, the study of a single animal species will not be sufficient to address overarching ecological and evolutionary questions. Consequently, as the field of neuroethology moves in the direction of understanding and incorporating the roles of multimodal signals for behavioral decision-making (i.e., visual, olfactory, gustatory, mechanosensory, and auditory cues), similarly, neuroethology is also beginning to examine a multitude of closely related animal species for evolutionary comparisons of morphology, behavior, and adaptation[13–15], which can help identify the selective pressures that drive these changes in sensory systems and neural development or neural plasticity.

One of the original genetic model organisms, the vinegar fly, *Drosophila melanogaster*, has been a workhorse of advanced genetics for the last several decades. The advantage of this invertebrate model is attributed to its short generation time, ease of colony establishment in the laboratory, the huge diversity of available molecular and genetic tools, as well as the immense efforts toward the complete mapping of neural circuits for both the adult and the larvae of this one species[16–18]. However, the genus *Drosophila* also provides between 1200 and 1500 individual species, with an ecology spanning nearly every imaginable environment and host choice, from deserts to forests, from islands to mountains, and across incredibly unique or specialized food resources, such as the gills of land crabs, protein sources within bat guano, or otherwise toxic fruits;[10,15,19–21] therefore, the potential to transform an already powerful model organism from a singular species into an entire genus is now possible due to the recent advances in cellular and genetic tools for examining the complex neurological mechanisms of natural behavior in novel, non-model species. Moreover, the expansion from a single species into an entire genus affords scientists the opportunity to address larger ecological, developmental, and evolutionary questions

using the full gamut of molecular and genetic tools that have already been generated for *D. melanogaster*. Research into non-*melanogaster* species is already well underway, with researchers beginning to highlight individual species, often selecting those based on economic impact or behavioral specialization[22–27], with studies now also including CRISPR-cas9, the powerful gene editing tool, such as the studies in *D. suzukii*, *D. subobscura*, *D. simulans*, and *D. pseudoobscura*[28–31].

An emerging integrative field of the biological study, called ecological evolutionary developmental biology, or more commonly known as eco-evo-devo, focuses on the underlying interactions between an organism's environment, its genes, as well as its development in regard to how these three factors shape evolutionary trends and help create a map or framework for better understanding and predicting speciation[32–35]. The field of eco-evo-devo is built on the premise that evolution is animal development controlled by ecological and environmental forces. Thus with the above-mentioned factors in mind, one of the goals of the present study is to encourage the expansion of the *D. melanogaster* model to become the *Drosophila* system, and thereby encompass a broader array of species within this genus for comparative, ecological research into what drives the evolution of the nervous system.

Based on the many examples from the animal kingdom as well as our previous observations from a number of Drosophilid species[27,36], we set out to test the hypothesis that sensory systems occupy a restricted niche in the nervous system of these flies, where relative size and energy allocation prevents one sense from expanding without having an effect on another. Also, as an entry to creating a larger ecological and evolutionary framework for this genus of flies, our study samples a wide, phylogenetic array of 62 different species within the genus *Drosophila*, and begins to analyze both host navigation and mate selection or courtship with regard specifically toward visual and olfactory sensory modalities. This study includes investigation at the periphery, such as morphometrics of the antenna and compound eye, as well as measurements within the antennal lobe (AL), optic lobe (OL), and the central brain for each selected species. This phylogenetic comparative approach allows for a more precise study of adaptation, and making these interspecific comparisons allows us to assess the general rules governing evolutionary phenomena via observations of repeated, independent evolutionary events within a group of organisms.

In our study, we identify a consistent, inverse resource allocation between vision and olfaction across these 62 species, and we use a combination of phylogenetic, phenotypic as well as developmental data in order to examine the evolutionary pressures and constraints underlying this potential tradeoff between two critically important sensory structures in regard to both host navigation and mate selection.

## Results

**Phylogeny, species selection, and general morphometrics.** An array of 62 species within the Dipteran family Drosophilidae were selected to span the diversity contained within the genus *Drosophila* (Fig. 1a, b). This genus of flies covers a multitude of hosts and host ranges, including examples such as rotten fruits, cacti, flowers, tree sap, and mushrooms. Each species was measured for a number of physical metrics, including body size, head size, eye surface area, and the surface area of the third antennal segment (the funiculus) (Supplementary Figure 1A). In general, there was a huge variety of physical sizes noted within this single genus of flies, providing much more variability in absolute or overall size between species than we initially anticipated. Not surprisingly, as fly species increased in either body or head size, eye surface area and funiculus surface area

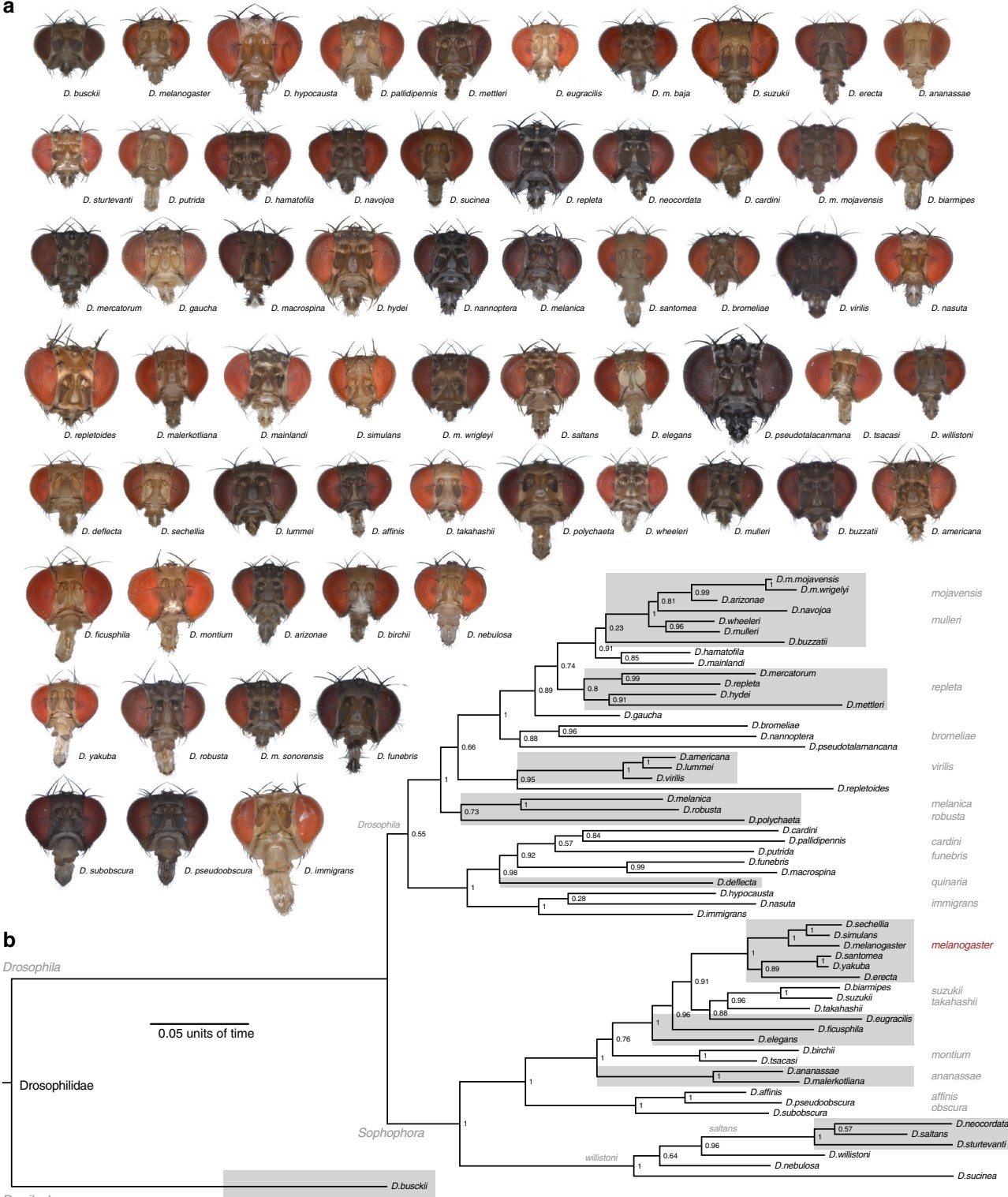

**Fig. 1** Frontal head images of all tested *Drosophila* species and their associated phylogeny. **a** Frontal view of the head of all 62 species, illustrating the diversity in overall size, as well as in the variance of the visual and olfactory sensory systems across this genus. Also worth noting is the disparity in pigmentation that extends across the whole head, including the antenna and the compound eye. **b** Phylogeny of 59 species of *Drosophila* where genetic material was available for use in this study (*D. montium* and two subspecies of *D. mojavensis* are missing). Species were selected to span the width of subgroups and represent the genetic diversity within this genus of insect. Some species are denoted with gray boxes to provide more visual separation between subgroups. (Data are provided at https://doi.org/10.17617/3.1D)

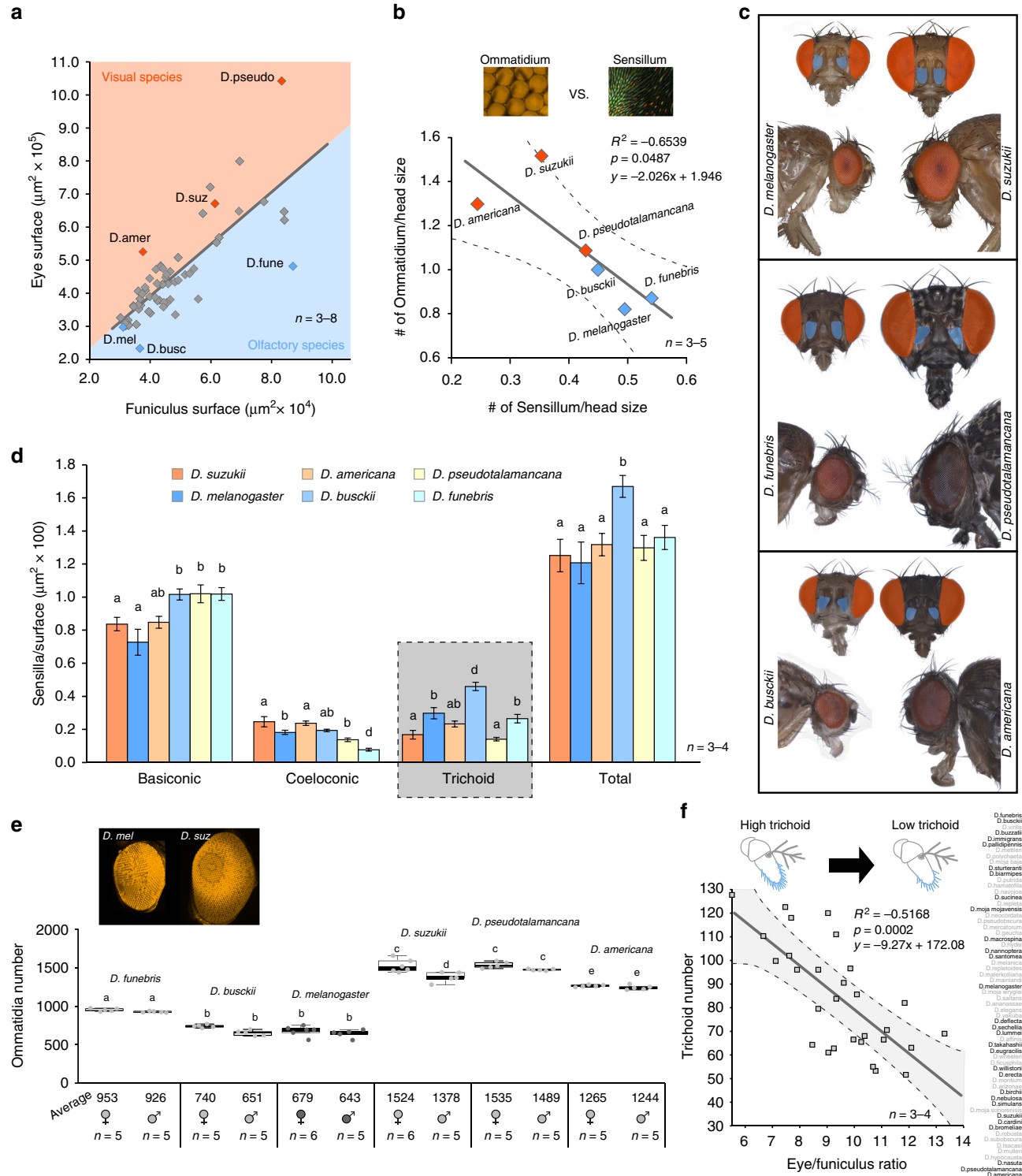

both increased as well, with head size always having a tighter positive correlation than body size for both eye and antennal metrics (Supplementary Figure 1). However, there was also quite a bit of variability in these sensory structures, both among similar body sizes and between flies with similar eye or funiculus sizes (Supplementary Figure 1). Here, we found that the eye and funiculus surface area scale isometrically with respect to both the body and head measurements (Supplementary Figure 1H); moreover, that the variance in these two sensory systems could not be explained by the absolute size of a species.

**Ommatidium and sensillum comparisons among main species.** For more in-depth comparison, we next sought to compare the sensory regions associated with visual and olfactory stimuli (Fig. 2a), and while again there was a general trend across the 62 species that larger insects had both larger eye surface area and larger funiculus surface area, there was still significant variability between these two sensory systems that was not explained by body or head size alone (Supplementary Figure 1H, I). From our robust array of species, we selected six Drosophilids for a more in-depth analysis of their sensory structures (Fig. 2a). These six

**Fig. 2** External comparison of visual and olfactory system. Red color signifies vision or predicted visual bias, while blue indicates olfaction or potential olfactory bias. **a** All 62 species measured for eye and funiculus surface area, where six species were selected for additional measurements. These flies were selected to compare species with similar antennal surface area but contrasting eye sizes (e.g., *D. pseudotalamancana* and *D. funebris*, or *D. americana* and *D. busckii*) or species with similar eye size but contrasting antennal sizes (e.g., *D. americana* and *D. funebris*). We also selected two well-established species, *D. melanogaster* and *D. suzukii*, for an additional comparison and points of reference. **b** Inverse correlation between ommatidium number and sensillum number when corrected for head size from six species of *Drosophila*, suggesting a possible tradeoff between these sensory systems at the periphery. **c** All species were photographed for more detailed measurements of eye and antennal features across several frontal and lateral views. Highlighted in blue are the antennal surface area, and in red, the eye surface area. **d** Shown are the sensillum density metrics taken from stacked lambda mode scans (maximum intensity projections) of the anterior portion of the antenna for all six species examined, identifying strong differences for example in trichoid sensillum density, where potentially olfactory biased species (in blue) showed the significantly larger trichoid densities. Error bars represent standard deviation. **e** Ommatidium counts from each species, which illustrates the large differences in visual capabilities across this genus of fly, with some species having 2–3 times larger eyes. Boxplots represent the median (bold black line), quartiles (boxes), as well as the confidence intervals (whiskers). **d**, **e** Means with the same letter are not significantly different from each other (ANOVA with Tukey–Kramer multiple comparison test). **f** Expanded study to include additional species (that were selected using stratified random sampling), where we show that trichoids are consistently and inversely correlated with increasing eye-to funiculus ratio across the entire genus. (Data are provided at https://doi.org/10.17617/3.1D)

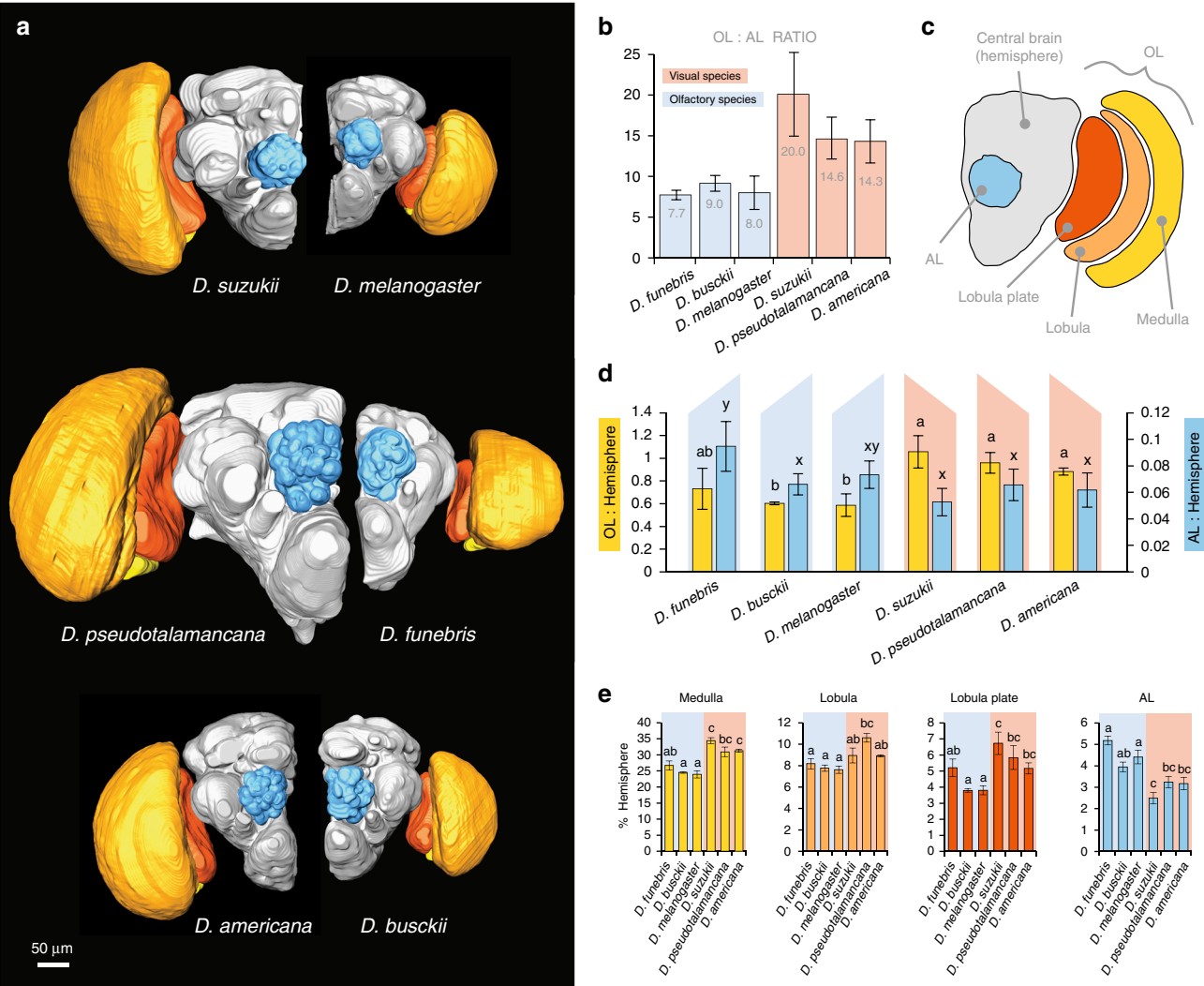

**Fig. 3** Three-dimensional reconstructions of the visual and olfactory neuropils in six *Drosophila*. Red to yellow (warm) color signifies vision or visual bias, while blue indicates olfaction or olfactory species. **a** Whole brain reconstructions, highlighting visual (yellow to red) and olfactory (blue) regions, with central brain in gray. **b** The optic lobe (OL) to antennal lobe (AL) ratio for each species, showing the division between olfactory and visual bias among species. **c** Diagram of all measured volumes for comparison between species. **d** Relative sizes of OL (yellow) and AL (blue) as compared to the central brain, where the data show an inverse correlation between visual or olfactory investment. **e** Separate regions of OL and AL that were measured as a percentage of the central brain to provide a comparable value between insects of differing absolute size, again highlighting that brain regions mirror external measurements of visual or olfactory size bias. **d**, **e** Means with the same letter are not significantly different from each other (ANOVA with Tukey–Kramer multiple comparison test). Error bars represent standard deviation. (Data are provided at https://doi.org/10.17617/3.1D)

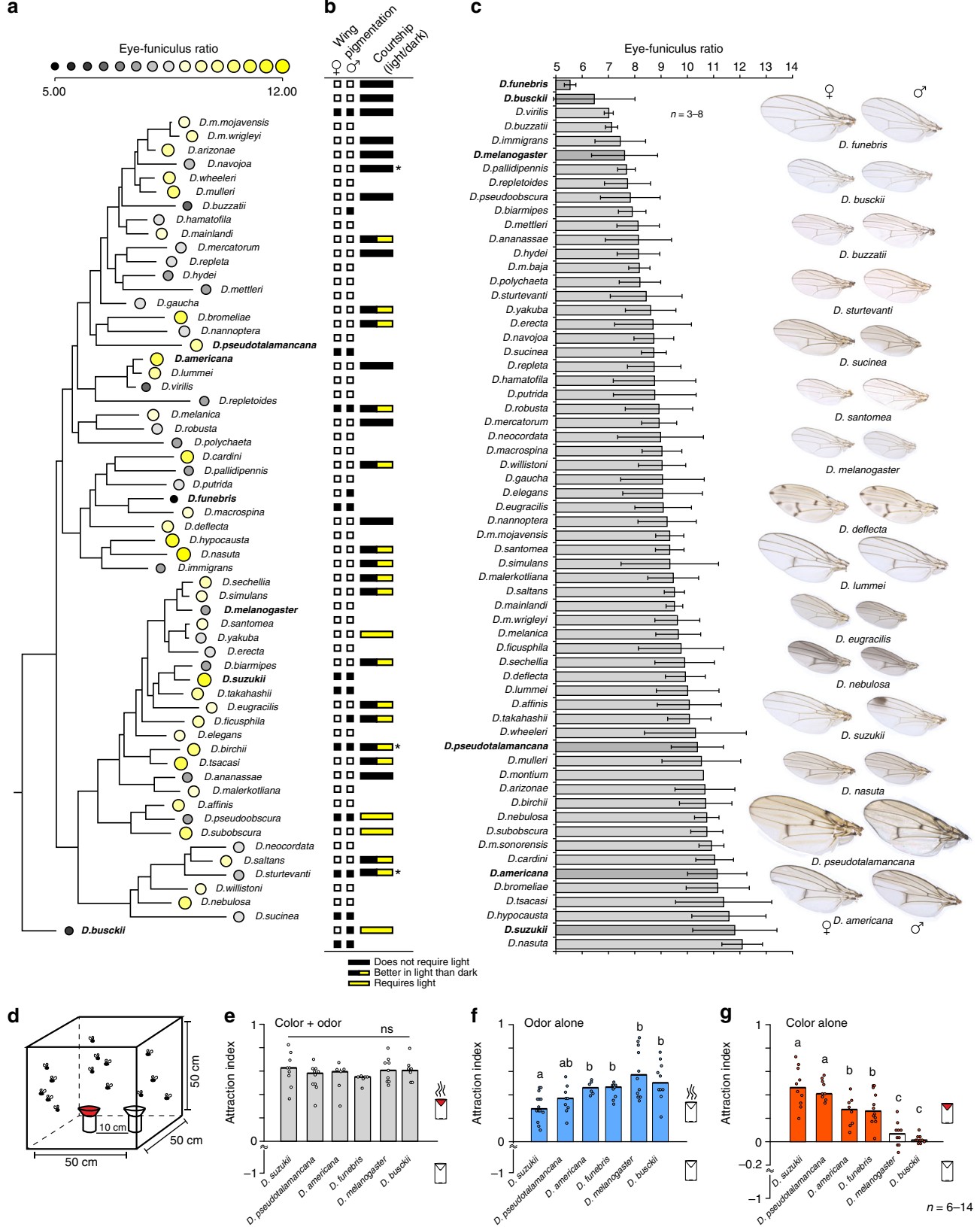

species were selected as either having similar funiculus size, but disparate eye size (i.e., *D. americana* and *D. busckii*; *D. pseudotalamancana*, and *D. funebris*), or vice versa (e.g., *D. americana* and *D. funebris*) (Fig. 2a). We also included *D. melanogaster*, given its prevalence in this genus as a model organism, and we

included *D. suzukii*, as it has risen to become both an important invasive species for agricultural research as well as an important model for evolutionary neuroethology.

We were interested in documenting any drastic differences in sensory structures beyond surface area (Fig. 2a, c), and we next

**Fig. 4** Host navigation and courtship differences across *Drosophila*. **a** Molecular phylogeny for 59 species that includes the eye-to-funiculus trait (EF ratio), which is visualized by both dot size and color. Two statistical tests (Blomberg K and Pagel's lambda) reveal that this sensory trait is not strongly supported by the phylogeny (K = 0.478, $p = 0.041$; $\lambda = 7.102e^{-05}$, $p = 1$). We note large variance within subgroups, and across habitat or ecological niche. **b** There was a significant correlation between both male/female wing pigmentation and EF ratio after phylogenetic correction ($p = 0.043$ and $p = 0.026$, respectively), suggesting that larger eyes correlate with pigmentation, which is not explained by phylogeny. Also shown are courtship values for mating pairs within light/dark environments, where light-based courtship is strongly correlated with larger EF ratio after phylogenetic correction ($p = 2.406e-07$), suggesting larger eye ratios correlate with visual mating. Asterisk indicates new data from this study. All other data from refs. [81–92]. **c** All 62 species arranged according to EF ratio, with wing pigmentation examples (standard deviation shown). **d** Diagram of behavioral assay used to test navigation of each species towards visual and olfactory objects. **e–g** Attraction indices for each species when stimuli were presented **e** together, **f** with odor alone, or **g** with visual target alone. While all species perform equally well when both odor and visual object are presented together, we observe a trend in behavioral preference where larger-eyed species perform more poorly in navigation towards odor objects when presented alone, but better towards visual objects, and vice versa for relative antennal size. (Data are provided at https://doi.org/10.17617/3.1D)

pursued additional metrics for visual and olfactory signal reception by quantifying sensillum and ommatidium number. Interestingly, the trend between visual and olfactory sensory structures was inversed among these six flies when we corrected for absolute head size (Fig. 2b), where large ommatidium counts in a fly species seemed to correspond with reduced sensillum counts, and vice versa. We also examined whether antennal surface area alone was a predictor of specific sensillum types, but surface area did not always predict the number of sensilla (Supplementary Figure 2G). In regard to olfaction, while these six species differed greatly in their absolute size, we discovered striking similarities in the density of sensilla found on either the anterior surface or the whole antennae (Fig. 2d; Supplementary Figure 2E, F). While both basiconic and coeloconic counts were roughly similar in their density, the largest difference between the species was in the number of trichoid sensilla (which have been shown to house sensory neurons detecting pheromone compounds[26,37,38]) (Fig. 2d). These trichoid differences were also apparent when we compared the absolute sensillum counts between species (Supplementary Figure 2D–F). Trichoids also varied in length and curvature. In addition to olfaction, we examined visual capabilities of each of these six species by counting the visual receptors or ommatidia (Fig. 2e; Supplementary Figure 2A–C, H), and again we noted large differences between these selected species, where ommatidia number was proportional to our previous measures of eye surface area. In order to further test the hypothesis that a tradeoff occurs between visual and olfactory sensory systems, we expanded our evolutionary comparison beyond these six examples to include additional species across the phylogeny (which were selected using stratified random sampling in order to represent as many subgroups as possible). Here, as before, we observed a significant inverse correlation between trichoid number and the eye-to-funiculus ratio (EF ratio) (Fig. 2f), where again, trichoid numbers were not correlated with antennal surface area or antennal size (Supplementary Figure 2G).

**Neuroanatomy of visual and olfactory sensory circuitry.** Given the disparity in external sensory morphology between our six species, we next sought to compare neuroanatomical metrics for the primary visual and olfactory processing centers within the brain (Fig. 3; Supplementary Figure 3). The species with the enlarged compound eyes also had a much larger OL relative to the AL, while the species with enlarged antenna had a relatively smaller OL (Fig. 3a, b). This matched our metrics related to external anatomy, suggesting as we predicted for example, that larger eyes correlates with larger OL volume. In order to account for differences in absolute size between each species, we used the central brain as a means to generate a weighed value for both OL and AL comparison (Fig. 3c–e). While it was not surprising that larger eyes or larger antennae matched with a larger brain region

associated with these sensory structures, we started to see a pattern where an increase or an exaggeration of one sensory structure correlated with a relative reduction in the other. For example, that while *D. suzukii* has a much larger (OL:AL) ratio or (OL: central brain) ratio when compared with *D. melanogaster* (Fig. 3b, d), at the same time *D. suzukii* also had a significantly smaller (AL:central brain) ratio by comparison (Fig. 3d). This trend is true for each of the other reconstructions and species comparisons. We also assessed the selected six *Drosophila* species in regard to subunits of the OL, including the medulla, lobula, and lobula plate, where again we saw a similar pattern of a significant increase in size for each subunit of the OL in larger-eyed species; moreover, that the medulla represented the largest increase relative to central brain volume (Fig. 3e; Supplementary Figure 3G). Here, we also documented again that the AL of the larger-eyed species was relatively smaller when compared with larger antennal species, as expressed by a ratio to central brain volume (Fig. 3e). While these six species varied in their absolute sizes (Supplementary Figure 3A–G), we noted that the central brain relative to the whole brain was consistent in size across all tested species (Supplementary Figure 3E), thus a relative comparison of OL or AL to the central brain within each species gave a consistent measure or weighted value for comparison.

**Phylogenetic correction of traits of interest.** To examine whether the phylogeny of our species could account for the variations, that we measured in the eye and antenna, we compared the EF ratio trait to all relatives within the genus (Fig. 4a). Here, we utilized two independent statistical tests of phylogenetic signal, including the Blomberg K value and Pagel's lambda (K = 0.478; $p = 0.041$; $\lambda = 7.102e^{-05}$; $p = 1$), where we assess phylogenetic signal to indicate the tendency for closely related species to resemble each other more than a random species selected from the tree. Here, we found that both statistical measures agree that this phenotypic trait (EF ratio) is not strongly supported by the phylogeny, where a K value less than one indicates that variation is larger within subgroups than between subgroups (Fig. 4a). Thus, while we considered phylogenetic associations as a driver of trait variation, we did not find a relationship between phylogeny and trait variation. In addition, we noted that eye and antennal size diverge repeatedly throughout the genus and were not predicted by known ecology or shared habitats (e.g., EF ratio was not correlated with cactus-feeding or desert-living species; Fig. 4a); however, more ecological data are still needed for a multitude of species to discern the role ecology plays in the observed sensory variation.

**Behavioral effects of sensory bias between species.** Given the trends and correlations we observed in our in-depth analyses of six species, and in order to assess potential behavioral courtship

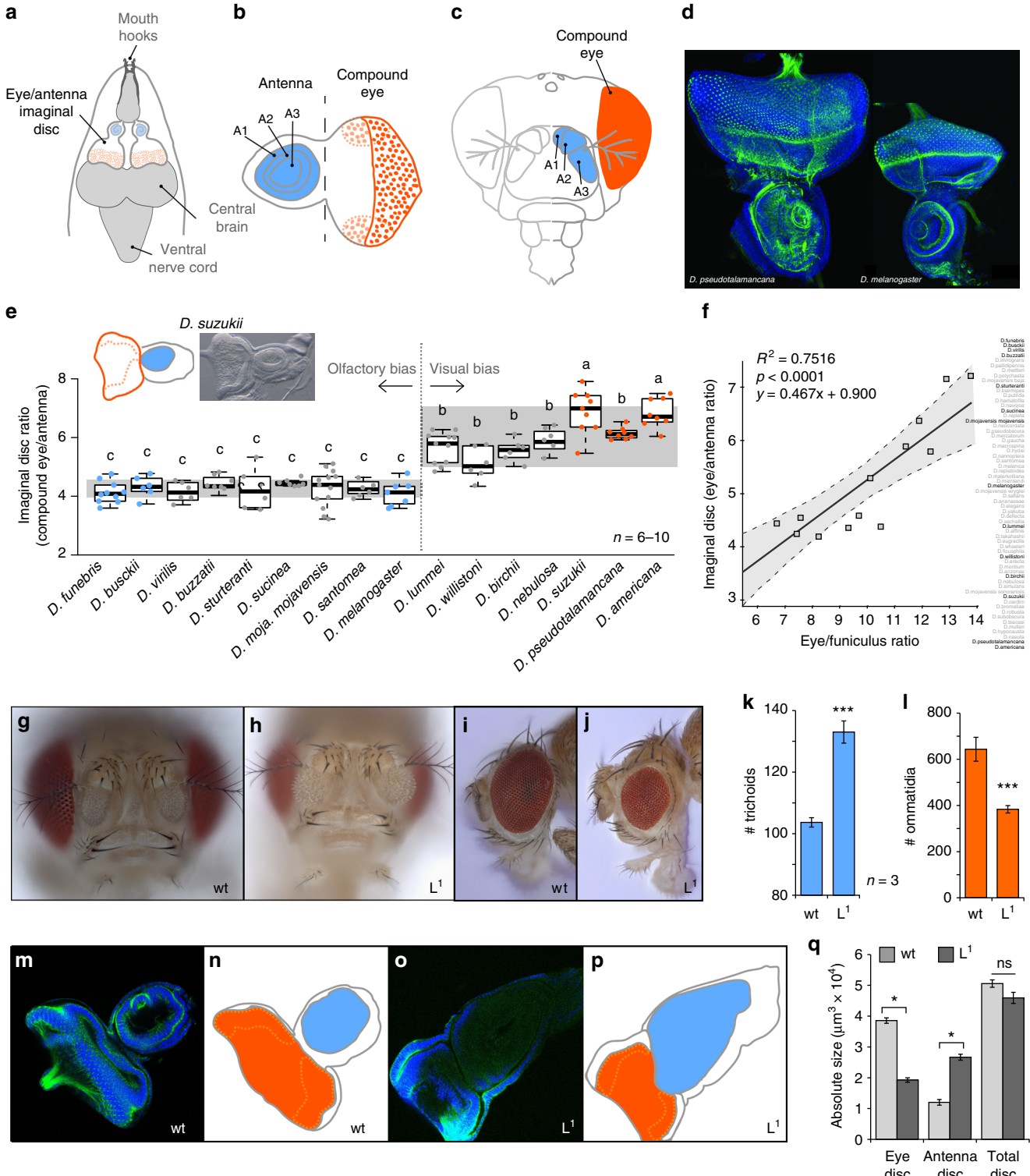

implications from the size variance of visual and olfactory sensory systems, we wanted to expand our comparative model to include all 62 species in our study (Fig. 4b, c). Here, we arranged all 62 species in regard to their EF ratio, as provided by measures of the surface area of each sensory structure, with smaller values indicating relatively large antennae, and bigger EF ratio values indicating a larger compound eye relative to the antenna (Fig. 4c). Photographs of wings from males and females were taken and used to provide information about wing spots or pigmentation for

each species that was tested (Fig. 4b, c), and we also used previous literature to assess whether each species is influenced by light (lux intensity) during courtship or whether light is required for successful mating to occur (Fig. 4b). There was a significant correlation between female wing pigmentation and EF ratio after phylogenetic correction ($p = 0.0429$) (Supplementary Figure 3H, I). In addition, there was a significant correlation between male pigmentation and EF ratio after phylogenetic correction ($p = 0.0256$); therefore, because there was a correlation between wing

**Fig. 5** Tradeoffs and developmental constraints. Red color signifies vision or visual bias, while blue indicates olfaction or olfactory species. **a–c** Diagrams of a single imaginal disc from larval development that gives rise to two separate adult structures, namely the eye and the antenna. **d** Two part staining (Hoechst & Phalloidin) of *Drosophila* species to visualize differences in absolute size of imaginal discs, highlighting the need for a ratio of eye to antenna for comparisons between species. **e** Imaginal disc ratios (eye to antenna) across each tested species where two groups were noted, olfactory biased and visually biased. Means with the same letter are not significantly different from each other (ANOVA with Tukey–Kramer multiple comparison test). Boxplots represent the median (bold black line), quartiles (boxes), as well as the confidence intervals (whiskers). **f** The significant correlation between larval imaginal disc measurements per species and the EF ratio from adult flies. **g–j** Eye and antennal mutants were compared to wild-type flies for both ommatidium and trichoid numbers. **k**, **l** From the mutants we screened, a single mutant, Lobe[1], displays increased trichoids and decreased ommatidia compared the the wild-type. An asterisk denotes statistical significance between two groups (*$p \leq 0.05$, ***$p \leq 0.001$; T test). **m–p** Eye–antennal imaginal disc comparisons between wild-type and Lobe[1] mutant, visualizing the tradeoff between visual (red) and olfactory (blue) development. **q** Measurements show that while the total size of the imaginal disc is the same between wild-type and mutant, that the proportion of eye and antenna are inversely correlated, suggesting a developmental constraint between these two sensory systems. (*$p \leq 0.05$, ***$p \leq 0.001$; T test) (Data are provided at https://doi.org/10.17617/3.1D)

pigmentation and EF ratio when we include the phylogenetic correction, the correlation between these two traits has no phylogenetic signal (i.e., the covariance of the residuals for the EF ratio and wing pigmentation regression do not follow phylogenetic signal). From the analyses of the light/dark courtship data in regard to EF ratio, we found these traits were strongly correlated both before phylogenetic correction ($p < 0.0001$) as well as after the correction based on relatedness of the species ($p = 2.406e-07$) (Supplementary Figure 3H, I). Thus in summary, it appears that proportionally larger eye size provides a potential visual bias in courtship that is associated with light-enhanced mating success. Moreover, we show that species with larger EF ratios (and thus those species with relatively larger eye size) were significantly more likely to possess wing pigmentation, and have significantly more successful copulation in light conditions (or display light-dependent courtship), perhaps as part of a successful visual display. However, due to the paucity of natural history for most species, additional work is needed to address all species-specific mating behaviors within this genus, including for example, pheromone-related courtship (or pheromone-related olfaction) in larger antennal species that display light-independent courtship.

As we had established a consistent difference between the visual and olfactory senses of the six species in regard to external and internal neuroanatomy as well as courtship, we wanted to next test if there was also any behavioral relevance to these sensory structure differences in regard to host navigation (Fig. 4d–g; Supplementary Figure 4A–D). When we combined visual and olfactory stimuli, all six species performed equally well in trap assays, including tests with several different olfactory cues, such as vinegar, blueberry, and strawberry (Fig. 4e; Supplementary Figure 4A). However, when we tested the olfactory stimuli alone, without any visual target, we observed a biased trend in that larger-eyed species navigated more poorly than larger-antennal flies (Fig. 4f), suggesting an olfactory advantage to large antennal species toward the odor object alone. The opposite phenomenon occurred when we tested visual stimuli in the absence of an odor source, where larger-eyed species performed significantly better than those species with enlarged antennae (Fig. 4g); moreover, we caught almost no flies from the larger antennal species using color alone. We also tested for species differences in their preference toward specific colors, with red and black being the most consistently attractive to all species, regardless of behavioral assay, but with *D. suzukii* also being attracted to green (Supplementary Figure 4A, B). However, this may be in part due to differences in contrast detection. Interestingly, *D. suzukii* was also more attracted to the combination of blue when presented with odor from blueberry, which may be linked to this species being reared for dozens of generations on this food source in our laboratory, and additional work will be required to test this combinatorial bias (Supplementary Figure 4A). In order to compare visible qualities of each

color used, we generated a diffuse reflection gradient for each visual stimulus, to confirm the primary visible wavelength associated with each color we used in this study (Supplementary Figure 4C). We also confirmed the reliance on visual stimulus for host navigation by repeating a trial in either full light and complete darkness (Supplementary Figure 4D). Here, for example, *D. melanogaster*, a large antenna, olfactory-driven species, navigated equally well toward an odor source regardless of light conditions (Supplementary Figure 4D). However, in the same experimental design, *D. suzukii*, a large eye, potentially more visual species, performed as well as *D. melanogaster* toward an odor source in the dark, but roughly split capture with the visual stimulus and the odor source when in light conditions. In this case, as all species were still able to locate a host source successfully using a single-stimulus type (i.e., odor object in the dark), it would appear that the difference in size of a sensory structure indicates an innate preference or behavioral bias for certain navigational cues, but that both sensory systems still work well. Although again, visual and olfactory stimuli worked optimally in tandem, or when the two stimuli were in agreement in regard to the location of the host (Fig. 4e). Future work should examine the behavioral response of each species when the visual and olfactory objects are not in spatial congruence in regard to the location of the host or food source.

**Evolutionary development of visual and olfactory structures.** Although insect development is a complicated and delicate process under strict genetic control, the process by which *D. melanogaster* undergoes development has been relatively well elucidated. In general, there are 19 imaginal discs from the *Drosophila* larvae, each of which gives rise to a different adult structure (Supplementary Figure 6A); however, there is only one disc that gives rise to several separate adult structures, namely the eye–antennal imaginal disc (Fig. 5a–d). Here, a single larval developmental structure generates primarily both the eye and the antenna for the adult fly (Fig. 5b, c). With this in mind, we next examined the relative ratio of the two sides of this imaginal disc, including both the eye and antennal portions across a multitude of species (Fig. 5e). Although species varied in egg to pupal developmental time, by dissecting the tissues from late third instar larvae (wandering phase; Supplementary Figure 7), we could generate consistent ratios for each species during the same time window of development (Supplementary Figure 6B, C). To confirm these measurements, we used two stains (Hoechst & Phalloidin) in order to more closely monitor areas separating these two portions of the same developmental disc in each new non-*melanogaster* species (Fig. 5d). By using a ratio between the two parts of the same imaginal disc, we could account for any issues during the comparison of species that differed drastically in absolute size, for example between *D. pseudotalamancana* and *D.*

*melanogaster* (Fig. 5d). Using the data taken from a multitude of *Drosophila* species, we could identify essentially two main groups or two common ratios, either antennal biased or visually biased (Fig. 5e). This developmental data matched very well with the previously established external metrics taken from the compound eye and antennal surface areas, and thus further support the theory that there is a tight link between the imaginal disc size for the eye and antenna in comparison with the corresponding adult structures (Fig. 5f). This data again provide evidence for an inverse resource allocation between the eye and the antenna during development, as these two sensory structures would essentially be competing for the same resources within a single disc (Supplementary Figure 6D).

**Genetic constraints on vision and olfaction**. While we could not further examine the role development plays in non-*melanogaster* species of Drosophilidae, we could in fact, examine established genetic lines within *D. melanogaster* for either eye or antennal mutations (Fig. 5g–q). In these experiments, we used previously identified mutations for either eye or antennal development in *D. melanogaster*, and analyzed both of these adult sensory structures in order to test our hypothesis that there is a tradeoff or inverse resource allocation (Fig. 5g–q; Supplementary Figure 6E–G). Here, we counted trichoid sensilla and individual ommatidia from each mutant line in order to assess any potential candidate genes that match the phenotype we observed in the wild-type species (Fig. 5g–l; Supplementary Figure 6E–G)). Although some fly mutants have been previously published for either visual or olfactory abnormalities, most lines have not to our knowledge ever been examined for both sensory structures within a single mutant. While not an exhaustive screen of all possible gene candidates in *Drosophila* development, we did uncover a single-mutant allele in our screen that appeared to have a similar tradeoff between visual and olfactory sensory structures to that observed across the genus, more specifically, Lobe[1] (L[1]), which has a significant reduction in the number of ommatidia while possessing a significant increase in the number of trichoid sensilla present on the funiculus (Fig. 5k, l), something that was consistent with the observations from wild types. This mutant has a reduced eye size, which has been previously published;[39–41] however, the alteration leading to increased antennal size (enlargement of all three segments) and the increase in trichoid sensillum number has not been previously described for this mutant (Fig. 5g–l).

In order to further test our hypothesis that the imaginal disc provides the framework for an inverse resource allocation based on the sharing of a single disc for two adult sensory structures, we next sought to examine the imaginal disc of this L[1] mutant in regard to eye and antennal ratio (Fig. 5m–p). Here, we observed that the Lobe[1] mutant has a marked reduction in the portion of this developmental disc that gives rise to the compound eye (Fig. 5o, p), while also showing a marked increase in the portion that gives rise to the antennal segments. When we measured the two portions of the developmental disc for both wild-type and mutant, we discovered that there was no significant difference in the total size of these imaginal discs (Fig. 5q), but rather that the proportion of the disc dedicated to each sensory structure had shifted in the mutant from the eye to the antenna (Fig. 5q). Thus, this new data lends additional support to our previous observation that a tradeoff might occur between visual and olfactory sensory systems, in this case during development, and that this inverse resource allocation is perhaps necessitated by the sharing of a single larval structure. Thus, for example, in order for the antennal region to increase in Lobe[1], there is necessarily a decrease in eye size to compensate. Recently, a preprint[31] has addressed this same developmental mechanism, and has proposed a similar tradeoff hypothesis by comparing two *Drosophila* species using CRISPR mutants, where they conclude that a single amino-acid shift can alter the functional timing of a gene, and explain the natural variation between eye and antenna during larval development. However, more research is needed to address whether this same developmental constraint can dictate the inverse correlation between visual and olfactory sensory systems that we have observed in all tested *Drosophila* species.

## Discussion

In this study, we provide large-scale evidence for an inverse relationship between visual and olfactory anatomical investment across this genus of Drosophilid flies. The potential tradeoff seems to stem from a theoretically restricted resource allocation between the eye and antenna during larval development, which is linked to a single shared structure giving rise to both adult sensory systems (Fig. 5d–i). It remains to be seen whether this push–pull between the eye and antennal region of the imaginal disc is under similar genetic control in all non-*melanogaster* species; however, our study and a recent preprint[31] provide evidence that a simple mutation can mirror inverse variation in ommatidia and sensilla numbers for *D. melanogaster*, something which is consistent with our observations of repeated, independent evolutionary events across this genus of fly in regard to visual and olfactory divergence.

Investment in an exaggerated sensory structure might be costly[42], thus prominent structures often result in a tradeoff with another trait to minimize energetic costs[43–47]. Tradeoffs can occur across populations or between species within a single subfamily or genus, and each different sensory structure often has differing ecological and environmental pressures acting upon it[48,49]. An example from vertebrates of a similar tradeoff hypothesis examines trichromatic color vision in primates[50], where researchers found that primates with heightened color vision also had a higher number of olfactory pseudogenes or non-functional gene mutations. In order to test this pseudo-gene argument, we also examined the olfactory genes from many *Drosophila* species using previously published data on OR, GR, IR genes, and their associated pseudogenes across 14 members of Drosophilidae (Supplementary Figure 1J)[51], but we did not find any meaningful correlation between olfactory pseudogenes and eye size or visual enhancement. However, it is possible that gene expression levels differ between *Drosophila* species, either across rhodopsin types or other visual pigmentation genes, or perhaps across olfactory-related genes. For example, while the most-studied *Drosophila* species have roughly the same diversity of chemosensory genes and ommatidium types[51,52], different olfactory receptor ratios exist across basiconic or trichoid sensillum types, where variation in olfactory receptor expression is often associated with specialization[10,25,26]. This was the case in *D. sechellia*, where this species has similar olfactory gene diversity (or number of chemosensory genes) when compared with *D. melanogaster*, but vastly different expression levels of a few specific receptors. Additional research is required to assess this type of expression-level comparison for visual and olfactory genes between a wider array of *Drosophila* species, as it is not clear if fly species with increases in ommatidia or sensilla numbers represent a uniform increase across receptor types. It is also important to mention that there are some limitations in our extrapolation to true wild-type insects due to the usage of stock center or laboratory flies, but we anticipate that our findings will extend to natural populations as well.

From an ecological point of view, we considered mate-finding and host navigation when examining sensory systems in *Drosophila*. Both of these behaviors have been shown to rely heavily on visual and olfactory inputs in several species that have previously been investigated. For example, wing pigmentation has been extensively studied in *Drosophila*[53–56], although never before in correlation with olfactory function such as pheromone detection (Fig. 4b, c). The removal of pigmentation heavily influences sexual selection and courtship, thus further confirming the importance of visual cues during courtship in spotted wing *Drosophila* as well as in the visual courtship of other animals[57,58]. In addition, it was recently shown that *D. subobscura*, which requires light for courtship success[59,60], has enhanced fruitless-labeled gene expression and circuitry that maps to the OL, unlike *D. melanogaster*, where courtship is light-independent[29]. Moreover, that study also highlighted fruitless-labeled visual enhancement into the lobula and lobula plate of *D. subobscura*, a specific increase in brain volume which we also show in all three of our visually biased species examples (Fig. 3e). Another well-studied example of courtship and incipient speciation is the diverging populations of *D. mojavensis*[22–24], where our data again show that the largest divergence is found between the closest relatives and geographically overlapping subspecies, suggesting character displacement as an additional driving force for the observed differences in visual and olfactory investment (Fig. 4a, c). In fact, the vast majority of *Drosophila* species we tested show the largest differences within a species clade or subgroup (e.g., *D. virilis* vs. *D. americana*; *D. biarmpies* vs. *D. suzukii*; *D. pseudoobscura* vs. *D. subobscura*), where courtship, mate selection, and host competition pressures are potentially highest, and perhaps driving repeated speciation events that favor either visual or olfactory bias to differentiate the species' niche (Fig. 4a, c). Although recent work has examined differences in the visual and olfactory systems of *D. melanogaster* and *D. pseudoobscura*[31], we do not feel this is a good direct comparison, given the poor phylogenetic connection between these more distantly related species (17–30 million years apart), and that other pairings would perhaps better tackle the genetic, ecological, and evolutionary pressures that underpin this sensory tradeoff (e.g., that *D. subobscura* or *D. affinis* would be a better comparison for *D. pseudoobscura*, while *D. simulans* or *D. sechellia* would be a better comparison for *D. melanogaster*). Thus, we conclude that the correlations and model provided by our study, including eye size and wing pigmentation as well as light-dependent courtship, match with previous publications from the *Drosophila* genus and our study provides a large dataset for further testing. In addition, our data continue to strongly support the theory that visual investment and OL increases mirror the behavioral priority of vision for courtship and/or host navigation in those species with larger EF ratios and wing pigmentation (Fig. 4b, c; Supplementary Figure 3H, I).

Although additional work is required to confirm any differences in pheromone production or increased olfactory courtship reliance in species with larger antennal ratios, our data already support the inverse investment between the eye and antenna in regard to copulation based on the number of trichoid sensilla versus ommatidia (Fig. 2b, d, f; Supplementary Figure 2 E–G). Moreover, within the suzukii subgroup, it has been well established that *D. suzukii* produces very low amounts of the male pheromone known as cis-vaccenyl acetate (cVA; detected by trichoid at1, and Or67d) and that this species has a greatly reduced glomerular volume within the AL for this odor[26]. The previous research matches our findings here that *D. suzukii* flies have a reduced total number of trichoids, and in addition, that these flies instead possess an enlarged compound eye that is 2.5 times larger than in *D. melanogaster*. Similarly, *D. biarmipes*, the

closest relative of *D. suzukii*, has also been previously studied and shown to have a large amount of cVA production, which is opposite to *D. suzukii*[36]. In the present study, we also found a correspondingly higher number of trichoid sensilla for *D. biarmipes* when compared with *D. suzukii*, even given the smaller overall size of *D. biarmipes*, matching a potential tradeoff between olfactory and visual investment between close relatives for courtship, again suggesting character displacement as a potential means of speciation or divergence (Fig. 4a, c).

Resource allocations have been well documented within other insects, such as in courting scarab beetles, where there is an inverse correlation of investment between physical horn size for fighting and sperm production for increasing the likelihood of paternity[61]. Examples of visual and olfactory variation have also been recently documented in other insects, such as in Lepidoptera, where nocturnal and diurnal species within the Sphingidae family of hawk moths vary widely in morphological investment toward either eye or antennal structures, as well as in their relative OL and AL sizes;[62] however, while a tradeoff between these sensory systems has not been previously proposed, these studies have shown by comparing two hawk moth species that relative brain structure increases match behavioral preferences, with diurnal species having enlarged visual centers and visual preferences, and nocturnal species having enlarged olfactory centers with olfactory behavioral preferences. Moreover, that these sensory brain measurements can be used to explain and predict differences in the importance or priority of these two senses (vision and olfaction) for host navigation. In these studies of Lepidopteran neuropils, it can be inferred from the data that investment in vision is perhaps associated with a relative decrease in olfactory processing centers, and vice versa, both for host-finding and migration, suggesting that perhaps an insect species cannot increase both sensory systems[62–64]. It has also been shown recently that a potential tradeoff might also occur between diurnal and nocturnal dung beetle species[65], where there was a difference across the two examined species between visual and olfactory brain regions based on circadian rhythm or daily activity patterns. Here, the diurnal species have a larger OL and are more visual, while the nocturnal species relies more on olfaction as well as possessing an enlarged AL. Another insect example of visual variation exists across Formicidae, where different ant species, or even different castes members within a species, have differing investment in vision depending on their ecological roles within the colony or depending on the amount of time they spend underground[66,67]. In addition, more distant insect relatives have been compared across visual brain structures[68], where the visual centers from Mantodea, Blattodea and Orthoptera were addressed for their anatomical similarities and differences. Although some of these latter studies did not address olfactory centers for relative comparison between both vision and olfaction, each example lends support to the hypothesis that all insects potentially demonstrate a tradeoff in sensory systems. However, additional work is still required in more orders of insects to assess this tradeoff hypothesis and the evolutionary pressures that lead to these potential compromises between sensory structures.

In many insect examples, the differential investment in OL or AL was linked to differences in activity (diurnal and nocturnal). These differences in circadian rhythm are not as well studied in all non-*melanogaster* species, and the timing of both courtship and host-seeking behaviors are not known for all species. However, in the *Drosophila* species that have been examined, they all share a similar crepuscular activity cycle, thus it is unlikely that differences in visual and olfactory sensory systems in *Drosophila* arise from nocturnal versus diurnal activity[60,69]. Additionally, tradeoffs between visual and olfactory signaling have been long

recognized in plant species, especially between odorous nectar or visual floral displays that are used in order to attract insect pollinators[70]. The difference in plants is evident where you have a visually large and distinct floral petal arrangement, but with reduced smell or reward. In contrast, other plants have little in the way of visual attraction, but utilize sweet nectar rewards or strong, pungent odor plumes to draw in olfactory-driven pollinators[71–73]. These plants examples again highlight potential differences across insect pollinators, such as hymenopterans and dipterans, where the plant takes advantage of insects that favor either visual or olfactory stimuli for host navigation, but perhaps not both sensory modalities[73]. It is possible in these cases that vision could assist some Drosophild species in finding their preferred plant hosts (i.e., flowers, or fruit ripening within leaves or tree canopies), although the paucity of ecological information for most species within this genus has made this impossible to examine so far.

In summary, our assessment of the genus *Drosophila* supports the hypothesis that the visual sensory system expands consistently at the expense of structures related to olfaction, and vice versa. In addition, we provide robust evidence that the inverse correlation observed between visual and olfactory sensory systems occurs repeatedly within the family Drosophilidae, and we conclude that our theory of a tradeoff is consistent with all observed patterns, and perhaps is necessitated by a developmental constraint. Moreover, while additional research is required to address the specific molecular genetic mechanism(s) that control this observed phenomenon across the entire genus, the data provided herein generate a solid foundation to continue to test this sensory tradeoff hypothesis in the future. By using a large subset of close relatives within one genus of Dipterans and creating an extensive overview of their visual and olfactory systems, including a robust molecular phylogeny, we were able to generate a finely tuned evolutionary framework, and we provide the first step in establishing a larger model system to encompass dozens of *Drosophila* species for additional study beyond *D. melanogaster* and its subgroup. In the end, we have also started to build evidence about the pressures and general rules governing developmental, ecological, and evolutionary phenomena related to differences in neuroanatomy and behavior across all insects, where the data provided support previous research as well as encourages new ideas and new avenues for the study of speciation, specialization, and the evolution of the nervous system.

## Methods

**Fly stocks.** All wild-type species, stock numbers, and rearing diets are in Supplementary Table 1. Unless otherwise noted, all fly stocks were maintained on standard diet (normal food) at 25 °C with a 12 h light/dark cycle in 70% humidity. Stock population density was controlled by using 20–25 females per vial. Mutants lines included oc[1] (ocelliless; Bloomington #2291), ar[1] (arista-less; Bloomington #210), Antp (antennapedia; Bloomington #2235), Dll (distal-less; Bloomington #3306), Diap[1] (thread; Bloomington #618), L[1] (lobe; Bloomington #318), gl[1] (glass; Bloomington #506), and gla[1] (glazed; Bloomington #1951). Stocks were maintained according to previous publications[74], and for all behavioral experiments we used 2–7 -day-old flies of both sexes.

**External morphometrics from head and body.** For each fly species or mutant line, 3–8 females were photographed using a Zeiss AXIO microscope, including lateral, dorsal, and frontal views. Flies of the 62 wild-types were dispatched using pure ethyl acetate (MERCK, Germany, Darmstadt). Lateral body (40×), dissected frontal head (128×), and dissected antenna views (180×) were acquired as focal stacks on an AXIO Zoom V.16 (ZEISS, Germany, Oberkochen) with a 0.5x PlanApo Z objective (ZEISS, Germany, Oberkochen). The resulting stacks were compiled to extended focus images in Helicon Focus 6 (Helicon Soft, Dominica) using the pyramid method. Based on the extended focus images, we measured body length (abdominal tip to antennal tip), head width (between eye margins), eye width, and eye height, as well as funiculus width and length, all measurements are

in μm (Supplementary Figure 1A). Assuming the eye as a full ellipsoid, we calculated the 3D surface based on the average eye width and half eye height as the ellipsoid radius (r), and used the formula $[4 \times (\pi) \times r^2]$ for the area of a sphere, then dividing the result by 2 to generate the eye surface area as a half-ellipsoid for each species. Calculations for the funiculus surface used its half-length and half-width as radius for the 3D ellipsoid surface area. Accounting for the proximal connection between funiculus and pedicel, we subtracted the circular base area, and then calculated with the funiculus width. In addition, we compared these calculations with previous publications for available species[52,75] in order to confirm that our metrics were similar, and while some of our estimates were low relative to other publications, they were consistent across replicates within each species. All raw measurements are available with the online library, as are the stock photos for all replicates (https://doi.org/10.17617/3.1D; 01 Species Images; Excel tables).

In order to test the validity of the usage of ratios for our comparisons made between visual and olfaction sensory systems, we have provided a statistical assessment of allometry (including a multiple regression analysis). First, we found that the eye and funiculus surface area measurements scale isometrically with respect to the measurements taken from the body and the head. Thus, we feel it continues to make sense to use the EF ratio as our primary trait, given that there is no real allometry in our data. Moreover, we show that neither body size ($p = 0.294$) nor head size ($p = 0.590$) significantly correlate with this EF ratio trait (Supplementary Figure 1H), and we have plotted the analyses of the residual variance (Supplementary Figure 1H). Last, we have also conducted a multiple regression analysis (using the EF ratio, eye, funiculus, body, and head measurements from all 62 species), and indeed again, the EF ratio does not correlate with body or head size in this multiple regression ($p = 0.354$ and $p = 0.295$, respectively). Overall, we continue to feel that we can safely maintain the usage of ratios, as the EF trait does not simply scale allometrically with body or head size, and these statistical tests again strengthen and further support our interpretations of the data that an inverse correlation exists between these sensory modalities that is not reflective of absolute body size. In addition, an online copy of the curated R scripts is available, including all measurements used to test allometry and to perform the multiple regressions (https://doi.org/10.17617/3.1D; 12Allometry).

**Ommatidium measurements.** In order to count ommatidia, the compound eye of each species was dissected and mounted on slides in water using a coverslip, and then photographed using a confocal microscope (Fig. 2e). A total of 5–6 individuals per species were used, and counts were done manually using ImageJ (Fiji) software tools (Supplementary Figure 2A). Diameters of single ommatidia were also assessed (Supplementary Figure 2B, C), with most species having roughly similar size.

**Sensillum counts.** Three different individuals from each species were anesthetized with $CO_2$, and their antennae were dissected. After removal, antennae were dipped into phosphate buffer (0.1 M pH, 7.3) with 5% Triton-X (Sigma-Aldrich) and they were washed in phosphate buffer and embedded in VectaShield (Vector Laboratories) between two cover slips[11]. To visualize the anterior surface of the antennae, lambda scans were obtained via confocal laser scanning microscopy (Zeiss LSM 880; Carl Zeiss) using a 40x water immersion objective (W Plan-Apochromat 40×/ 1.0 DIC M27; Carl Zeiss) in combination with the internal Argon 488 -nm laser (LASOS) and the 405 -nm Laser diode (Carl Zeiss). The broad emission spectrum of the samples auto-fluorescence was detected with the quasar detector (Carl Zeiss). Thereby images with 32 separate channels (each with a range of 9.7 nm) are generated simultaneously (Supplementary Figure 2D). To visually support the following sensilla quantification, lambda scans were post processed using the linear un-mixing technique (Carl Zeiss; http://zeiss-campus.magnet.fsu.edu/articles/spectralimaging/introduction.html). This technique enables the determination and separation of spectral profiles for every pixel and assigns each pixel, according to its spectral profile, to a manually defined spectral group. Three spectral groups were defined by selecting reference points in each stack (diameter 5 pixels) using the ZEN software (Carl Zeiss). This technique enables reassignment of one color for each group to a region (or group of pixels) that would otherwise appear as mixed color, and therefore supports visual separation of olfactory sensilla from other structures as well as the characterization of different sensillum types, due to structural differences (e.g., between trichoid, coeloconic, and basiconic shapes) that cause distinct emission spectra in their auto-fluorescence.

The sensillum quantification was done with the cell counting plugin (https://imagej.nih.gov/ij/plugins/cell-counter.html) in ImageJ (Fiji). Linear unmixed lambda stacks were visualized as a composite of all three channels and sensilla were manually counted by going through the stack. Each sensillum was assigned to one group (trichoid, basiconic, and coeloconic) and marked separately, and then each group was summed in the end.

Sensilla density of each anterior surface side was calculated as follows:

$$\text{Sensilla density} = \frac{\text{Sensilla number}}{\frac{1}{2}\text{funiculus surface}(\mu m^2)} \quad (1)$$

For trichoid sensillum counts of the other 24 species, counts were done manually for either the anterior or posterior or for both sides of the antennal surface. Counts were conducted with images from a Zeiss AXIO microscope under bright-field light, using arista up single sensillum recording preparations for each

insect that was examined (Supplementary Figure 5A, B), as this was the best preparation for viewing and counting trichoid sensilla[37]. A total of 3–6 individuals were counted per species, and where possible, these totals were compared with previous scanning electron microscopy (SEM) images, or lambda scans, or the previously published counts from the available species.

**Phylogeny of *Drosophila* species.** Species were initially selected, ordered, and arranged to include close relatives in pairs or triplicates for each major subgroup within the genus. Our initial molecular phylogeny search consisted of 16 mitochondrial and nuclear genes that were identified and used previously for studies of Drosophilidae[76,77]. However, many of these sequences were partial, or from older literature, while in addition, some genes had representation in only a few species. Therefore, we replaced much of the previously published data with the newer sequences that are currently available in public sources such as GenBank and Flybase repositories, with new sequences being either complete or longer in length than those that were previously published. In particular, no segments of the same gene in a species have been combined, as had been done in previous publications. We retrieved only the nucleotide coding sequence (CDS) regions of protein-coding genes, as well as the nucleotides for non-coding ribosomal RNA genes. In cases where mitochondrion genomes were available (bold after species names), then all the target mitochondrion genes sequences were retrieved from the same genome data. Moreover, in cases where the sourced data contained multiple genes, the specific region of the target gene sequence is given. After we assessed each individual gene, we generated trees for each gene individually, and ultimately narrowed our list from 16 down to 5 genes for concatenation (ADH-1, Amyrel, NADH-2, NADH4, and NADH4L). Raw molecular data, including sequences and accession numbers, are available at https://doi.org/10.17617/3.1D; 02 Molecular Phylogeny and in Supplementary Data 1.

For phylogenetic tree construction, we used available sequences from 59 Drosophila species drawn from the *Sophophora* and *Drosophila* clades, including *D. busckii* as an out group in the *Dorsilopha* clade of this genus. We assessed the dataset for each of the 16 gene families for quality in terms of representation or coverage across the sampled species, completeness of sequence length, the nucleotide multiple sequence alignment conservation, as well as the ability of each gene to reconstruct the phylogeny of the species represented (for individual phylogenetic trees see https://doi.org/10.17617/3.1D; 02 Molecular Phylogeny). This assessment enabled us to also determine the sequential order for concatenating the genes. Our final concatenated dataset were comprising two nuclear protein coding genes, amylase related (AmyRel) and alcohol dehydrogenase subunit 1 (ADH-1), as well as three mitochondrion genes, NADH: ubiquinone oxireductase subunit 2, −4, and −4L (NADH-2, NADH-4, and NADH-4L). We excluded non-coding mitochondrion genes for the reason that they individually failed to reconstruct the phylogenetic tree, as the sequences were often partial, had biased representation across the species, or failed to reproduce a consistent phylogeny, though we still include them for future reference in the online library (https://doi.org/10.17617/3.1D; 02 Molecular Phylogeny). The final dataset consisted of 229519 bp data points, in 59 concatenated sequences. The sequences were multiply aligned using a MAFFT tool with L-INS-I parameters, with 10000 bootstrap (Kato & Toh, 2008) and the final tree was reconstructed using maximum-likelihood approach with GTR+G+I model of nucleotide substitution and 1000 non-parametric bootstrapping, re-sampling of 10 initial random trees in Fasttree program. We did not partition the concatenated gene sets in this analysis. All emanating trees were visualized, and rendered using Figtree v.1.4.2.

Using this newly created phylogeny, we analyzed in two different ways the phylogenetic relationship for the eye–funiculus trait that we had generated for each species. First, we tested the Blomberg K value (K = 0.478; $p = 0.041$), where the K value being less than one suggests a lower phylogenetic signal than expected from Brownian motion; moreover, this low K value indicates that the variance is mostly within a given subgroup, and not between subgroup clades. Here, we determine phylogenetic signal to indicate the tendency for closely related species to resemble each other more than a random species selected from the tree. Second, we tested the Pagel's lambda value ($\lambda = 7.102e^{-05}$; $p = 1$), where again, a $\lambda$ value that is not significantly different from zero indicates very little phylogenetic signal in this trait. Thus, given the consistency of these two different statistical measures, we determined that the eye–funiculus ratio is not strongly supported by the phylogenetic relationship of the species that we tested.

**3D reconstructions and neuropil measurements.** In order to assess neuroanatomy, the dissection of fly brains was carried out according to established practices[78]. The confocal scans were obtained using multiple photon confocal laser scanning microscopy (MPCLSM) (Zeiss laser scanning microscopy [LSM] 710 NLO confocal microscope; Carl Zeiss) using a 403 water immersion objective (W Plan-Apochromat 40×/1.0 DIC M27; Carl Zeiss) in combination with the internal Argon 488 (LASOS) and Helium-Neon 543 (Carl Zeiss) laser lines. Reconstruction of whole OLs and ALs was done using the segmentation software AMIRA version 5.5.0 (FEI Visualization Sciences Group). We analyzed scans of at least three specimens for each and reconstructed them in using the segmentation software AMIRA 5.5.0 (FEI Visualization Sciences Group). Using information on the voxel size from the laser scanning microscopy scans as well as the number of voxels labeled for each neuropil in AMIRA, we calculated the volume of the whole AL as

well as the individual sections of the OL and the central brain (where central brain values exclude the AL volume).

**Behavioral assays for visual and olfactory stimuli.** Trap experiments were performed as previously described for individual odors[27,36], but using white or colored paper cones as an entrance to the trap (as non-*melanogaster* adults were too large to enter pipette tips). We also used an additional 200 μl of light mineral oil (Sigma-Aldrich, 330779-1L) that was added to capture and drown flies upon entering to the paper cone trap, and to ensure they did not escape over the 24 h testing window. Trials were conducted with 30 adult flies (15 males, 15 female), and each species was run separately. All behavioral cone traps consisted of 60 -ml plastic containers (Rotilabo sterile screw cap, Carl Roth GmbH, EA77.1), with one trap used as a white control and the other containing a colored cone entrance (red) (Fig. 4a–d, Supplementary Figure 4A, D). In experiments with whole fruit, each fruit was placed individually into traps that were presented simultaneously, where the sides of the container were opaque to avoid any extra visual stimuli, and as before, a large arena was used (BugDorm-44545 F) (Fig. 4a; Supplementary Figure 4A, D). For Petri dish behavioral traps (Supplementary Figure 4B), color paper circles were cut out and placed onto standard 10 -cm Petri dishes, either with or without an odor source, where mineral oil was again used to capture flies that landed on the paper disks. A total of 60 adults (30 males, 30 females) were used per trial, with a 16 L:8D photoperiod during testing. All odor dilutions were prepared in hexane or water, and all behavioral trials were conducted with odors diluted to $10^{-3}$ unless otherwise noted. Statistics were performed using GraphPad InStat version 3.10 at both $\alpha = 0.05$ and $\alpha = 0.01$ levels. No differences were noted between the sexes in regard to behavior, and thus, the data were pooled.

**Color and wavelength measurements.** The measurement of the backward light scattering with directed reflection took place using a Lambda 950 spectrometer (Perkin Elmer). This device is suitable for measurements in the UV/VIS/NIR range from about 200 nm to 2500 nm. The measurement of each colored paper was conducted at discrete wavelengths in this range with a distance of 1 nm (Supplementary Figure 4C), which allows for the more discrete characterization of each color used (i.e., green reflected light between 480 and 580 nm, and was well within the expected range for this color).

**Wing pigmentation and light/dark courtship.** The wings from male and female adults from each species were dissected and mounted with a slide and coverslip, with images generated using a Zeiss AXIO microscope under bright field and transmitted light (Fig. 4e, f). Wing pigmentation was noted for males and females from all species (https://doi.org/10.17617/3.1D; 08 Wings), with examples shown for most wings with any spots or pattern, where there was a significant trend of wing pigmentation being correlated with larger eye species relative to antennal size (Fig. 4c; Supplementary Figure 3H). Previously published data for courtship that required light, or where courtship was better under light conditions (yellow bars in Fig. 4e) or where courtship was possible in the absence of light (black bars in Fig. 4e) are shown (Supplementary References), with new data denoted by an asterisk. Light-dependent courtship, as well as mating better in light conditions, was also correlated with larger eye size relative to the antenna, suggesting a connection between vision and visually-mediated courtship signals such as wing pigmentation (Supplementary Figure 3 I). For statistical measurements, we used the package caper (Comparative Analyses of Phylogenetics and Evolution in R)[79] as well as the packages ape (Analyses of Phylogenetics and Evolution) and phytools (Phylogenetic Tools for Comparative Biology) to perform phylogenetic generalized least squares (pgls) and employed Pagel's lambda, Blomberg K, and the Brownian model of phylogenetic relatedness, with the R-script available online. We chose the caper package as we were most comfortable with the way it handles missing data, for example during the analyses of light/dark courtship, where published behavioral data are missing for several species. For all three phenotypes (female wing pigmentation, male wing pigmentation and courtship in light-dark), the estimates of Pagel's lambda for the branch length transformation significantly deviate from a strict Brownian motion process model of phylogenetic relatedness (i.e., deviate from lambda = 1; for more details, please see R-script at doi.org/10.17617/3.1D; 02 Molecular Phylogeny).

**Staining of imaginal discs.** Fly species were selected using stratified random sampling in order to represent as many subgroups as possible. Third instar larva were allowed to self-clean for several minutes in 1 M phosphate-buffered saline (PBS) and then dissected in fresh PBS. In a first dissection step, the imaginal discs were kept attached to mouth hooks and central brain to add structural stability. This coarse dissection product was transferred into 0.5 -mL reaction tubes with fresh, cold 300 μL of 1 M PBS. The PBS was exchanged against cold 400 μL of fixative, and the tissue was incubated in the paraformaldehyde solution on ice for 35 min. Next, tissue samples were washed in cold 400 μL of 1 M PBS five times for 5 min each. After removal of the PBS, the dissection products were incubated in the blocking solution on ice for 45 min. Then the blocking solution (1 M PBS plus 7% normal goat serum) was replaced with the staining solution (blocking solution with 0.07% Hoechst and 1% Phalloidin 488) and samples were incubated on a rotator at 4 °C for 2 h. Subsequently, the tissue was washed again in cold 400 μL of 1 M PBS

 13

five times for 5 min each. In a fine dissection step, the imaginal discs were then freed from all other connected tissues, and then mounted on object slides using a drop of Entellan® (Merck, Darmstadt, Germany). Sections of the imaginal disc were measured in Fiji software, and ratios were generated of surface areas for the eye divided by the corresponding antennal surface area (Fig. 5h; Supplementary Figure 6C), with 6–14 replicates per species, always taken from third instar wandering phase larvae just prior to pupation (Supplementary Figure 7).

**Statistics and figure preparation**. Statistical analyses were conducted using GraphPad InStat 3 (https://www.graphpad.com/scientific-software/instat/) and R Project (https://www.r-project.org/), while figures were organized and prepared using R Studio, Microsoft Excel, and Adobe Illustrator CS5. Additional details concerning tests of allometry, multiple regression, and phylogenetic correction are contained within the publically available R scripts that are described below in the Code availability section.

**Reporting summary**. Further information on experimental design is available in the Nature Research Reporting Summary linked to this article.

**Code availability**. All scripts for R, including curation of what tests were conducted, as well as the raw data files used for each statistical analysis are available at DOI: 10.17617/3.1D [10.17617/3.1D] (see 02 Molecular Phylogeny; 12 Allometry)[80].

## Data availability
All data supporting the findings of this study, including methodology examples, raw images and z-stack scans, molecular sequences, accession numbers, statistical assessments as well as species information are all available through Edmond, the Open Access Data Repository of the Max Planck Society, https://doi.org/10.17617/3.1D [10.17617/3.1D][80].

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

## Acknowledgements

This research was supported through funding by the Max Planck Society. Genetic mutants used in this study were obtained from the Bloomington Drosophila Stock Center (NIH P40OD018537), and wild-type flies were obtained from the San Diego *Drosophila* Species Stock Center (now The National Drosophila Species Stock Center, Cornell University). We express our gratitude to S. Trautheim and her team for their technical support and guidance at MPI-CE. We thank Ibrahim Alali for his help with fly rearing and maintenance. Thank you to Dieter Gäbler from the Fraunhofer Institute for Applied Optics and Precision Engineering (IOF) for his support during the wavelength and reflectance measurements. We would also like to thank the research teams within the Department of Entomology at the University of Missouri, Division of Plant Sciences, and the scientists within the Department of Evolutionary Neuroethology, MPI-CE, in Jena, Germany, for their insights and comments.

## Author contributions

This study was built on an idea conceived by I.W.K., while V.G., B.S.H., and M.K. all contributed to the design of this study. V.G. and I.W.K. completed the images and measurements associated with body morphometrics and ommatidium metrics. V.G. handled all neuroanatomy measures as well as the 3D reconstructions. L.G. and I.W.K. worked on the sensillum counts, while L.G. completed the lambda scans for antennal descriptions. I.W.K., G.B., and B.A.B. conducted the behavioral trials. I.W.K. and S.K. performed the imaginal disc experiments and metrics, including labeling, staining as well as confocal scans, with S.L.L. and J.R. providing their expertise. I.W.K. and D.R.V. worked on the courtship and wing images, as well as the data analyses. G.F.O., I.W.K., and M.A.K. assessed and built the molecular phylogeny, where D.R.V. and G.K. completed the statistical analyses for phylogenetic correction. M.A.K. and I.W.K. selected, ordered, and maintained fly species. I.W.K. prepared the original paper and all figures, while I.W.K., B.S.H., and M.K. all contributed to the final manuscript and subsequent revisions.

## Additional information

**Competing interests:** The authors declare no competing interests.

