## [Peer Review File · Nature Communications]

Reviewers' Comments:

Reviewer #1:

Remarks to the Author:

This paper describes an inverse relationship between the resources allocated to visual versus olfactory anatomy among different species of *Drosophila*. Different allocations to these systems have been observed previously in species of other insect orders. However, the present manuscript describes them for *Drosophila*, which is of special interest because of the wealth of knowledge about some of these species and the potential for exploring this issue further.

The manuscript provides a large amount of data, providing measurements of many parameters, and it examines 62 species. In principle this study should make a very valuable contribution to the literature. Unfortunately, the manuscript contains a variety of flaws that make it difficult to assess.

There are many claims that are not adequately supported by statistics.

--For example, the Fig. 4F legend indicates that "larger eye ratios correlates strongly with wing pigmentation and the necessity of light and vision for courtship success." There are both examples and counterexamples of this correlation in the data that are shown.

--line 738 "slightly higher tendency" This is hard to judge.

Some of the conclusions are not easy to draw from the figures:

--"Spotted wing species took longer to make a choice." Fig. S4D does not show times.

--the trichoid sensilla are hard to see in Fig. S5B

--third instar larvae are hard to see in Fig. S7B (are they supposed to be shown in A?)

The figures are not numbered in either the figures themselves or in the titles of the Supplementary files. Within a figure or figure legend, some of the panels are misnumbered:

--line 666 should be G, not J

--Figure S2 has two panels labeled B

There are many graphs where the axes are not labeled clearly (Fig. 2B should be "Number of ommatidia") or the units are not clear (Fig. S1B). How are the areas of the blue and yellow regions defined in Fig. 3B? The legend to Fig. 2D-F should indicate what the green and red colors represent. What are aIR and dIR? The n values should be clearly specified in the figure legends, as should the sex of the flies that were examined. In some figures the variance is not shown.

In one figure, some of the labels are in German (Fig. S4).

How is the anterior side of the antenna defined? The shape of the antenna differs in different species. Do trichoid sensilla cover different portions of the antenna in different species?

The writing needs improvement in many places. To provide one example that needs clarification, in the paragraph beginning on line 313, "Therefore, at least in *Drosophila*, there is not a direct connection to the identity of olfactory genes in association with visual modification; however, it is also possible that expression levels differ between these species, either across rhodopsin types or other visual pigmentation genes, or perhaps in olfactory-related genes. For example, between different olfactory receptor ratios within basiconic or trichoid sensillum types, where variation in olfactory receptor expression often matches speciation."

Other matters:

Line 816 and line 856 – how well did the values agree with published values?

line 150: why is the trend between visual and olfactory sensory structures reversed when corrected for absolute head size?

Why does *D. biarmipes* have fewer trichoid sensilla and a smaller eye/funiculus ratio than *D. melanogaster* (Fig. S8)?

Fig. 2D-F – what happens when *D. americana* is compared directly to *D. funebris*?

In the behavioral experiments, would the odors that are used be expected to activate trichoid or basiconic sensilla?

The authors should discuss why they think anatomical parameters should correlate strongly with acuity or discrimination. The *C. elegans* olfactory system accomplishes a great deal with a vanishingly small number of neurons. Human color vision operates with three opsins. Through what mechanisms do more neurons impart improved function?

Reviewer #2:

Remarks to the Author:

In this ambitious work, Bill Hansson and colleagues explore the idea that sensory specialization in different *Drosophila* species is accompanied by a trade-off between the visual and olfactory systems. Through a systematic examination of the peripheral sensory organs and brain neuropils across 62 different drosophilid species, they suggest that there is an inverse relationship between the size and neural territory allocated to visual and olfactory processing. From behavioral data they propose that this sensory trade-off biases species to be specialized in either visual or olfactory tasks. Finally, the authors examine how developmental constraints may bias the size and structure of these sensory tissues.

Understanding how evolutionary selection works in concert with developmental mechanisms to shape the nervous system is an important question. This work represents a really admirable attempt to gain insight into some of the basic principles of how nervous systems evolve and is a refreshing push into examining neural architecture through the lens of evolution. However, fundamentally the work remains extremely descriptive. More importantly, I have several concerns about the experimental design and analysis.

1. A key element of their analysis is based on comparison of specific neural metrics, like the eye/funiculus ratio. However, it is not obvious whether these are the appropriate readout to examine the evolution of sensory systems. For example, given the regular 2D packing of similar ommatidia, the surface area of the eye would seem a good proxy for expanded sensory receptors. However, in the olfactory system, the 2D surface of the antenna may have nothing to do with how odor is perceived.

2. Why do the authors present some comparisons only pairwise between selected species? It makes it difficult to appreciate the global tradeoff derived from all species.

3. The authors use simple behavioral assays to support the idea that there is a correlation between sensory specialization at the neural and behavioral level. However, one assumption is that quantitative differences in eye size or antennal sensilla translate to better olfactory tracking or color discrimination. This of course is not necessarily true--the species-specific expansion of ommatidia may not be used to extend color vision but for higher visual acuity or motion detection. Likewise, the choice of odor stimuli

in the trap assay appears unrelated to the ecology of the species used and may not reflect quantitative differences in sensilla. For example *suzukii* and *melanogaster* are likely to be attracted to qualitatively different olfactory cues (fresh vs. rotting fruit) and so would be unlikely to equivalently respond to the same set of stimuli. Differences in behavior could therefore reflect differences in odor tuning not differences in sensilla number as proposed. Indeed, the expansion of olfactory biased species seems to reflect larger numbers of trichoid sensilla, which in *melanogaster*, are thought to be tuned to pheromone, not the food odors used in their assay. Olfactory expansion could therefore be purely related to differences in chemical communication between species.

4. The idea that wing patterns are prevalent in visual species and related to mate preferences seems an over-interpretation. Many species rely on vision but do not have patterned wings. With the exception of *suzukii*, few of the visually biased species have sexually dimorphic patterned wings making their role in courtship less clear.

5. The authors suggest that the tradeoff between vision and olfaction is necessitated by the sharing of a single developmental structure—the imaginal disc. This argument arises from the existence of a mutant (*Lobe*) where eye progenitor tissue is lost while antennal tissue is expanded. However, not discussed is the observation that other mutations that showed reduced eye size but did not affect trichoid number (*gl*, *gla*) and mutations affecting trichoid number did not show a corresponding increase in ommatidia number (*Dll*), leaving doubt as to whether the effect of *L* mutations on trichoid number are truly reflective of developmental trade offs between eye-antennal structures as opposed to a more specific phenotype relating to this mutation. Given the wealth of tools available in *melanogaster* that allow for the expansion or restriction of growth in specific cells, the test of their model using these mutations feels unconvincing. Furthermore, the author's model of a developmental trade off. If the model is that the shared origin of the eye and antennae creates a competition for the pool of undifferentiated progenitor cells each organ has to recruit from then wouldn't the other structures arising from the eye-antennal disc (i.e. the maxillary palps, ocelli, head capsule) show the same inverse relationship? Finally, it would be good to clarify the difference between atrophy of a sensory system that is not used and the inversely coordinated tradeoff proposed here. For example, if we assume that a species relies more heavily on vision, the size of its eyes may be increased while they rely less on other modalities and are therefore smaller. However, this does not necessarily arise from a constraint.

Reviewer #3:

Remarks to the Author:

My main concern is there is no clear description of how the data was analysed. Without understanding how the authors analysed the data it is difficult to comment in depth on their results. Of primary concern is whether the authors analysed their data within a phylogenetic framework? It is clear they have a phylogeny but were their correlations performed accounting for phylogeny? There are also methods such as ancestral trait reconstruction which may help the authors consider their results within an evolutionary framework. Given the authors are embarking on an evolutionary and ecological framework the authors need to be upfront about the limitations around the use of stock centre flies.

The authors are claiming a trade-off between eye size and funiculus surface area they need to tread a little carefully here. Whether the observed correlational between eye size and funiculus surface area represents a functional trade-off or selection, cannot be distinguished across the species. It could possibly be that selection rarely operates in the direction of big eyes big antennae and these structures are costly to maintain such that relaxed selection would result in a reduction in size/use of these structures. The mutant work, only in *D. melanogaster*, lends further support for a functional trade-off but I found this section to be poorly describe and thus difficult to interpret.

It is clear the authors have done a lot of work but there were an overwhelming number of figures and I found it difficult to understand how it all came together.

Line 67 This is a vague statement, it is unclear how an inter-specific comparison will shed light on the field of neurotheology.

Roughly how long has each species been maintained under laboratory conditions? Can the authors please add this to their supplementary table.

Line 130 I assume the authors mean head size and funiculus are more strongly correlated than body size and funiculus size.

Line 275 There is no mention of candidate genes anywhere else in the manuscript. I am unsure how this section fits with their work as there is no clear methods for what they did.

Line 316 The authors have the data to ask some interesting questions about the origins and evolution of these structures across the *Drosophila* phylogeny but fail to shed any real light on evolutionary process across the phylogeny.

Line 321 The authors introduce new data in the discussion? There could also be a power issue here?

Methods

Did the authors control for density which would have clear implications for body size?

The methods section lacked any clear explanation for the purpose of their assays, making it difficult to determine why they did what they did. Simple sentences at the start such as To examine..... would greatly improve the readability and understanding/justification of their methods and conclusions. For example I found the methods for the behavioural assays confusing with no clear defined purpose for these assays. I am unsure to what end the authors were aiming to achieve with these assays.

How did the authors analyse their data? I can not seem to find this either in their methods or their supplementary materials.

Reviewer #4:

Remarks to the Author:

Keese et al present an analysis of investment in visual and olfactory sensory structures across a large number of *Drosophila*. Their results imply a trade-off between sensory modalities, and the authors suggest that this is evident in the external anatomy, neuroanatomy and neural development of these species.

There is a lot to admire about this paper. It contains a huge amount of work, combining different neuroethological approaches in a broad comparative context. It is a hugely ambitious study that, off the top of my head, is unique among the invertebrate literature, and is exactly the kind of comparative neurobiology that I think is needed in an era dominated by model organisms. I was wholly ready to absolutely love the paper. Unfortunately, I don't – yet! I do think there is an excellent paper here, but the current version has a number of issues pertaining to the data analysis and interpretation. As a result, I think the paper doesn't quite do the idea or data justice. However, these issues should be relatively straightforward to address and I anticipate that the revised manuscript can be much improved.

I'll try to explain the general issues I have, then provide some more minor, specific comments. The main issues are:

- 1) a lack of phylogenetic correction
- 2) use of ratios
- 3) inference of trade-offs

1) Phylogenetic correction: the introduction nicely sets up the idea of expanding the neuroethology approach across a phylogenetic scale to investigate evolutionary questions. However, evolutionary biologists recognise that species are non-independent due to the hierarchical nature of their phylogenetic relatedness. As such, interspecific data cannot be analysed using standard statistical methods. Phylogenetic non-independence is important as it can effect not only the significance of any regression analysis, for example, but also the slope and residual variance. Statistical approaches that remove, or correct, for the effects of phylogenetic non-independence are needed. There are several ways of doing so now, implemented in quite accessible packages such as BayesTraits, Phylotools, APE. Essentially all the statistical analyses in this paper should be re-done using these methods. This would include the regressions in the figures.

2) Use of ratios: Throughout the paper the authors use a ratio of two traits for comparisons of investment, and to correct for allometric effects caused by differences in body size or brain size, for example. This is very common in the comparative neurobiology literature but, in my opinion, is quite flawed. Using a ratio assumes your two variables scale isometrically with a slope of 1. This assumption does not hold in the majority of biological cases (see, for example, Voje 2016, *Am Nat* 187(1)). Without wanting to be patronising, I will try to explain why I think this is important. If your traits do not scale isometrically the slope of the regression will be significantly greater or less than one. This slope is interesting in itself as it can be used to infer some predictions about the constraints involved in the relationship between the two traits, but it also affects the inference of proportions. If two traits scale hypo-allometrically, with a slope of <1 , assuming isometric scaling will make it look like the Y variable in larger individuals/species are disproportionately small compared to smaller individuals/species. The converse is true for hyper-allometry. As such, ratios can produce misleading measures of relative size. Of course, the authors may argue ratios capture the percentage investment level from a mixed energy budget. However, this assumes the traits are free to vary in that way. If they scale non-isometrically, they may not be able to. The proportional size may therefore not reflect an investment trade-off, but may instead be determined by some functional scaling effect that is not related to energetics.

I've written a couple of small things about these effects in other context elsewhere; in the hope that these may be helpful, see: Montgomery 2013 *Brains Behav Evol* 82 (3); Montgomery & Mank 2016 *Mol Ecol* 25 (20). I would replace all the analyses with ratios with multiple regressions (in a phylogenetic context). Where the authors want to plot something like a ratio I'd perform a phylogenetic regression and plot the residuals. Of course, the alternative would be to test for isometry in the data – if your traits do scale isometrically the use of ratios is probably justified. However, even here there are problems - the other thing to point out is that when the authors do not have an independent variable (e.g. central brain size), so are comparing OL/AL or imaginal disc compound eye/antenna for example, there inevitably has to be a negative relationship as if one trait gets bigger the other one will 'appear' smaller, even if it has in fact stayed the same size. So, in sum I would not use ratios to infer trade-offs.

3) Inference of trade-offs: I'm not 100% on board with the conclusion that there is a trade-off between the sensory modalities for a couple of reasons. One is statistical, I'd like to confirm the negative relationship correcting for the issues above, but I imagine it will hold. The second is ecological. If I have followed the authors argument correctly I think they are saying the trade-off reflects competing demands of energetic investment (i.e. the flies have to 'choose' between visual

investment and olfactory investment). However, later the argument seems to morph into the trade-off existing due to negative genetic covariance (as suggested by the development/mutant lines). I don't think either is necessarily convincing. First, the alternative explanation is simply that these species evolved in different habitats and these habitats select for differential investment in each sensory modality. In some habitats visual information is more reliable – so selection favours increases in visual structures – but olfactory information is less reliable – so selection favours decreases in olfactory structures. In other habitats the opposite is true. In the discussion the authors use variation across Lepidoptera as another example of trade-offs between the senses. I actually think this better illustrates the role of ecology. Yes, OL and AL size are negatively correlated across Leps, but when you look at the detail the variation is much more subtle. For example, there are diurnal butterflies that invest in AL size to a similar extent as some nocturnal moths, while still investing in the big OLs typical of butterflies (see, for example Montgomery & Ott, 2015 J Comp Neural 523:869-891). So in my view, selection seems to be able to bring about increases in both sensory modalities, if it wants to (this is quite an adaptationalist interpretation, admittedly, but here the trade-off is about utility not development or energetics).

The developmental work presented in the manuscript does not convince me that this could not also be the case here. First, all the imaginal disc work shows is that development mirrors the final phenotype – surely this is to be expected? It doesn't really show that two areas of the imaginal disc can't evolve independently. Second, the mutant line work does provide a mechanism for linking the relative size of the two traits, but obviously does not rule out the possibility of there being other mechanisms that could permit independent evolution. To rule out an ecological explanation the authors would either have to show that there is no genetic variation in these species permitting independent evolution (very hard to do but you could take a quantitative genetics approach) or to incorporate ecological data in the analyses to see if this predicts the interspecific variation (perhaps also infeasible due to lack of data). So perhaps for now the authors can just discuss the alternative explanations in addition to their favoured interpretation. One analysis that may be possible to add would focus on intraspecific variation. If a genetic/developmental constrain is responsible for the trade-off you expect it to operate within individuals as well as between species. If the authors have large enough intra-specific datasets they could test whether or not the negative relationships are apparent within species, which may bolster their argument. If they are not consistently negative correlated, I'd probably interpret this as evidence against the current interpretation.

More specific points (many of which relate to the above):

Line 66: "the study of a single species will also often be insufficient to address... evolutionary questions" – I would say never! Unless you're studying inter-population differences and micro-evolutionary process. From a comparative perspective, one species is $n = 1$ so zero statistical/inferential power.

Line 129-133: here multiple regressions may be useful to test whether body size has any effect (also on line 140). I wasn't sure what 'zone 1' and 'zone 2' mean here (biologically). If the point is just to show there is variation you could estimate the regression parameters and plot the residual variance. As the variance is much reduced in panel C I would guess a lot of the effects relate to variation in selection shaping body size, rather than eye surface area.

Line 142: its not clear what you mean by species pairs – I tried to locate these 6 species on the phylogeny (which took a while - maybe you could label them as you have done for other comparisons) and (I think) they do not form phylogenetically independent pairs (of red/blue datapoints), which is what I assumed you meant. As such your regression will be affected by phylogenetic non-independence here as well. As the text is written now, *D. americana* also appears in both groupings – which must be an error since you say 'vice-versa'

Line 162-163: what about omatidium size? If surface area and number both vary you could calculate estimates for this. I think this would be interesting. If facet area is larger it may imply adaptations for photon capture, whereas increases in facet density may suggest selection for greater acuity (maybe?). You might also predict this would affect neuropil differently. For example, increasing facet number without increasing facet size should increase lamina/medulla volume. Whereas increasing facet size without increasing facet number might not as there would be the same number of cartridges.

Line 168-192: Why was the lamina not measured? Is it damaged in the dissections?

Line 176: when using central brain volume as a control, do you subtract AL volume?

Line 184: it looks like ME, LOB and LOP may vary somewhat independently of total OL. Is this not of interest? Total OL size will also be dominated by the effects of ME variation, so if LOB and LOP are varying independently this could be masked to some extent.

Line 189: "we note that the central brain hemisphere as part of the whole brain was consistent in size across all tested species" – I'm not really sure what panel E shows. If the values are percentages they seem very small? Also, if this was true then there would be no effect of correcting for central brain size as the denominator would shift numerators in a consistent way. So are you really just seeing effects of absolute size of the neuropil? If so, do these vary negatively? If not, is there any real trade-off?

Line 223: "based on our arrangement" – I don't get why the authors don't do any analyses here. An arrangement of data doesn't provide any evidence in support of their hypothesis. You could easily do a (phylogenetic) multiple regression with eye size and/or funiculus size with binary variables for wing pigmentation and/or courtship to test if these ecological traits explain variation in sensory investment. As it is, its just descriptive (fine, but not convincing).

Line 256: "species that differ drastically in absolute size" – I think this is a key point in absolute terms both traits are increasing in size/number. Using a ratio doesn't really correct for these allometric effects. As the analyses are presented I think it remains possible both are increasing independently, but to differing degrees.

Line 264: "structures would essentially be competing for the same resources within a single disc" – only if there are no developmental mechanisms that can promote localised proliferation/growth.

Lines 316-331: I find this paragraph unconvincing, in particular the correlations with the number of sensory genes. I'd remove it as it doesn't show much anyway (and suffers from phylogenetic issues). Its unlikely this idea holds up – for example in Leps, OL size varies hugely between nocturnal/diurnal species, but most species have 65-70 OR genes.

Lines 376-397: see comments above on Lep brains.

Figure 1: are the branch lengths meaningful? Panel A would make a great cover image!

Figure 3: can you add the sample size for each species (as in figure 2G). The pairing of species in panel D is also a bit random. For example, americana is more closely related to funebris and pseudo. than it is to busckii. Indeed the closest phylogenetic pairing (excluding melanogaster/suzukii) would be americana/pseudo.

Data: Perhaps I missed it but I didn't see any reference to data accessibility – can the raw data be made available?

I hope these comments are taken as constructively as they are intended! As I said, I think there is a great, exciting paper here once these issues are addressed. I'm happy for the authors to contact me if it would be helpful to discuss any of the issues further.

Reviewer #1 (Remarks to the Author):

This paper describes an inverse relationship between the resources allocated to visual versus olfactory anatomy among different species of *Drosophila*. Different allocations to these systems have been observed previously in species of other insect orders. However, the present manuscript describes them for *Drosophila*, which is of special interest because of the wealth of knowledge about some of these species and the potential for exploring this issue further.

The manuscript provides a large amount of data, providing measurements of many parameters, and it examines 62 species. In principle this study should make a very valuable contribution to the literature. Unfortunately, the manuscript contains a variety of flaws that make it difficult to assess.

There are many claims that are not adequately supported by statistics.

--For example, the Fig. 4F legend indicates that "larger eye ratios correlates strongly with wing pigmentation and the necessity of light and vision for courtship success." There are both examples and counterexamples of this correlation in the data that are shown.

We have now taken statistical measurements from this figure, including phylogenetic corrections pertaining to the relatedness of the species. We have drawn more suitable (and well-supported) conclusions from this reanalysis. Female wing spots significantly correlate with Eye-Funiculus ratio (EF ratio) after correcting for the phylogeny ($p = 0.04291$ for females); moreover, male wing pigmentation was significantly correlated with EF ratio after phylogenetic correction ($p = 0.02557$ for males).

Additionally, the light/dark courtship data does appear to be strongly correlated with EF ratio, both before and after phylogenetic correction ($p < 0.0001$, $p < 0.0001$; respectively).

In both cases, we have added statistical figures to Supplemental Figure 3 (H,I), and in the raw data files we provide, and we include all R-scripts used for these statistical analyses of wing pigmentation and courtship.

--line 738 "slightly higher tendency" This is hard to judge.

Agreed, and we have adjusted the text following the new statistical measurements.

Some of the conclusions are not easy to draw from the figures:

--"Spotted wing species took longer to make a choice." Fig. S4D does not show times.

Due to the sheer size of the dataset, we eventually elected to remove some figures, but did not manage to adjust the text where we had omitted some of the data. This has been corrected. Thank you for spotting this error in the text!

--the trichoid sensilla are hard to see in Fig. S5B

We were limited in file sizes for initial submission of figures, but have now provided a link for the raw data, including all image files that were used to calculate trichoid sensilla, for example. These images are composite scans to provide focal depth. We hope the sensilla are more clearly visible in the final version of the figures and in the supplemental raw data. Please see hosted DOI link to access raw image files.

--third instar larvae are hard to see in Fig. S7B (are they supposed to be shown in A?)

Thank you. We have fixed this issue in the new version of Fig. S7B.

The figures are not numbered in either the figures themselves or in the titles of the Supplementary files. Within a figure or figure legend, some of the panels are misnumbered:

--line 666 should be G, not J

--Figure S2 has two panels labeled B

Corrected! Thank you for your attention to detail; this was an oversight on our part! We have included the figure number and title in the name of each file we submitted, and we will confirm this with the editor, but that each attached figure file should be correctly labeled (e.g. Supplemental Figure 2). Thank you for bringing this to our attention.

There are many graphs where the axes are not labeled clearly (Fig. 2B should be "Number of ommatidia") or the units are not clear (Fig. S1B). Adjusted to meet your requests, we hope these new labels provide more clarification in the figure or the accompanying legend. How are the areas of the blue and yellow regions defined in Fig. 3B? Removed, as we agree, these were confusing. The legend to Fig. 2D-F should indicate what the green and red colors represent. We have moved these images to supplemental, and provide additional explanations in the figure legends (Fig. S2) as well as methods section. What are aIR and dIR? Antennal IRs and divergent IRs, but clarification added to text to mirror original publication from which the data was obtained. The n values should be clearly specified in the figure legends, as should the sex of the flies that were examined. We have added sample size, and added sex to legends or in the text for each data set. Trichoid numbers were assessed from males, all other data is from females. In some figures the variance is not shown. We have added standard deviation error bars to all figures, as well as the sample size.

In one figure, some of the labels are in German (Fig. S4). Thank you! An oversight on our part, and has now been corrected!

How is the anterior side of the antenna defined? Diagrams/Drawings have now been added to Fig. S2E in order to assist in clarification. The shape of the antenna differs in different species. Do trichoid sensilla cover different portions of the antenna in different species? While we did not measure this specifically, from observations of the different species, I can tell you that this appears to be correct; the regions of sensillum types are different between species. Density appears to be relatively conserved for most sensillum types, but the bounds for trichoids, for example, differs between species. Moreover, this observation appears to mirror the work done previously on *D. melanogaster* sensilla/ORNs based on their relative position within the eye-antennal imaginal disc (Chai et al., "Sensory neuron lineage mapping and manipulation in the *Drosophila* olfactory system", bioRxiv, 2018; Bayramli and Fuss, "Born to run: patterning the *Drosophila* olfactory system", 2012, *Developmental Cell*). Additional work would need to be done to compare antennal shape, developmental region, and patterns of expression for specific ORs across species of interest; but again, we feel your observation is correct that shape and coverage by sensillum types differs between the tested species.

The writing needs improvement in many places. To provide one example that needs clarification, in the paragraph beginning on line 313, "Therefore, at least in *Drosophila*, there is not a direct

connection to the identity of olfactory genes in association with visual modification; however, it is also possible that expression levels differ between these species, either across rhodopsin types or other visual pigmentation genes, or perhaps in olfactory-related genes. For example, between different olfactory receptor ratios within basiconic or trichoid sensillum types, where variation in olfactory receptor expression often matches speciation.”

We apologize, and we have sought to enhance and correct this example to be more clear and concise, as well as checked and edited other areas of the manuscript, and we hope this enhances the readability and clarity of the ideas and hypotheses that we try to convey.

Other matters:

Line 816 and line 856 – how well did the values agree with published values?

We added to the text some examples from previous publications that count ommatidia and sensilla. Our ommatidia counts and measurements match very well with existing literature that we cited, including Posnien et al (2012) and Arif et al (2013). We also compared our sensilla counts to other older literature, such as Stocker et al (2001), and again, our numbers match quite well with those that were previously published.

line 150: why is the trend between visual and olfactory sensory structures reversed when corrected for absolute head size?

We feel this is due to a developmental tradeoff, which becomes most apparent when you correct for absolute size differences between the 62 species.

Why does *D. biarmipes* have fewer trichoid sensilla and a smaller eye/funiculus ratio than *D. melanogaster* (Fig. S8)?

These types of examples are interesting, and further examination of this evolutionary topic needs to be addressed with additional research. Aspects to consider include: the comparison of relatedness, the habitat or ecology of these species, or other genetic constraints (perhaps limitations or bounds of absolute increase/decrease in structures). However, these two species are very close in regard to both EF ratio ($D_{bia} = 7.9029$, $D_{mel} = 7.6146$) as well as trichoid counts ($D_{bia} = 96$, $D_{mel} = 102$).

Fig. 2D-F – what happens when *D. americana* is compared directly to *D. funebris*?

We have added ANOVA statistics to better allow the direct comparison between species for which we have detailed data available. In this example, you can test more clearly these species in Fig.2D as well as SFig.2F which have been adjusted to examine these types of questions. Here we can see that while density for sensillum type is similar between these two species (Fig.2D), the absolute numbers are quite different (SFig.2F), where *D.funebris* has twice as many total sensilla as *D.americana*, mostly due to a doubling of basiconic and trichoid sensillum types, and a doubling of the antennal surface area.

In the behavioral experiments, would the odors that are used be expected to activate trichoid or basiconic sensilla?

For this study we tested 3 main food odors for behavior (Fig. 4E-G, SFig. 4A), and did not find a significant difference between these odors (vinegar, strawberry, blueberry), which we believe all activate mostly basiconic sensilla. I would highlight that we see a consistent response when the visual and olfactory cues are presented together for all tested species (Fig. 4E), as in this case, *Drosophila* are all attracted to vinegar, hence the common name of vinegar flies for this genera appears to be

accurate. A subsequent and unpublished study from our lab will continue to outline specifically the trichoid and trichoid-related odors from a similar set of 60+ species of *Drosophila*, which we expect to see published in the coming 6-12 months. As such, we have left out trichoid-specific odors and trichoid-specific behavior from the present study.

The authors should discuss why they think anatomical parameters should correlate strongly with acuity or discrimination. The *C. elegans* olfactory system accomplishes a great deal with a vanishingly small number of neurons. Human color vision operates with three opsins. Through what mechanisms do more neurons impart improved function?

Thank you, we have added a mention of this into the discussion. We assert that increases in type as well as neuronal number are highly influential for the olfactory system. Examples of OSN overexpression include species specialists such as *D.sechellia* and *D.erecta* (both overexpress Or22a and Or85b; Dekker et al. 2006 Current Biology; Linz et al. 2013 Proceedings of the Royal Society B), while another species, for example, has reduced expression of a key odorant receptor for courtship or aggregation, namely the reduction of Or67d (at1 sensillum) and therefore cVA detection in *D.suzukii*. In these examples, each is associated with behavioral shifts, where increases in expression across the first two species leads to increased sensitivity, tolerance and attraction, while reduction in expression of the latter species leads to reduced behavioral metrics associated with a specific aggregation and courtship pheromone. Again, these species examples possess the same olfactory receptor, but differing numbers of neurons associated with them produce large variation in behavior.

However, we concur, additional work is needed to compare specific ORNs across these species, as most of the work on non-*melanogaster* species has focused on presence/absence studies of OSNs at the genome level, and these studies lack functional descriptions, numerical or expression values, and also lack behavioral testing. It has also recently been demonstrated that the peripheral detection in *Drosophila simulans* of a specific chemical has its behavioral valence reversed, due to a wiring change in the AL and projection neurons (Seeholzer et al. 2018, Nature). Thus it will also be important in the future to assess not just odor detection at the periphery, but also the associated behavioral response or valence of key odors across a wider array of species within this genus, as evolution has many ways to create species-specific changes in neuronal function. Having said that, we feel this manuscript represents the first steps in pushing future research into more evolutionary and ecological territory, using dozens of *Drosophila* species as a model genus to support larger assessments and tests of general assumptions about evolution, such as how neuron type, expression level, and valence can influence olfactory function and host-shifts. The same can be said for visual receptors. Again, we have added some of your suggestions into the discussion, and thank you again for your comments!

Reviewer #2 (Remarks to the Author):

In this ambitious work, Bill Hansson and colleagues explore the idea that sensory specialization in different *Drosophila* species is accompanied by a trade-off between the visual and olfactory systems. Through a systematic examination of the peripheral sensory organs and brain neuropils across 62 different drosophilid species, they suggest that there is an inverse relationship between the size and neural territory allocated to visual and olfactory processing. From behavioral data they propose that this sensory trade-off biases species to be specialized in either visual or olfactory tasks. Finally, the authors examine how developmental constraints may bias the size and structure of these sensory tissues.

Understanding how evolutionary selection works in concert with developmental mechanisms to shape the nervous system is an important question. This work represents a really admirable attempt to gain insight into some of the basic principles of how nervous systems evolve and is a refreshing push into examining neural architecture through the lens of evolution. However, fundamentally the work remains extremely descriptive. More importantly, I have several concerns about the experimental design and analysis.

1. A key element of their analysis is based on comparison of specific neural metrics, like the eye/funiculus ratio. However, it is not obvious whether these are the appropriate readout to examine the evolution of sensory systems. For example, given the regular 2D packing of similar ommatidia, the surface area of the eye would seem a good proxy for expanded sensory receptors. However, in the olfactory system, the 2D surface of the antenna may have nothing to do with how odor is perceived.

We show that density of sensilla (and density of sensillum type) is pretty well conserved across the measured species, and that the major difference is the overall size or surface area of the antenna (where size plays the greatest part in predicting total sensilla number). Also, the antennal surface area seems to mirror changes in AL volume, with increase in antennal size matching the increase in AL size; however, as we did not address specific ORNs we cannot say whether or not this also matches an individual glomerulus volume (though we would predict from our data that this would be the case for specific ORNs). Moreover, we cannot say at this time whether antennal surface area changes odor perception in the brain, though we can say that larger antennal surface area does correlate with a trend towards increased olfactory driven behavior (Fig.4E-G). Thus in summary, we feel that funiculus surface area is a good proxy for odor detection and odor-driven behavior, though additional work is required for odor perception differences between species and between receptors.

2. Why do the authors present some comparisons only pairwise between selected species? It makes it difficult to appreciate the global tradeoff derived from all species.

Initially, we selected species in pairs for comparisons between similar eye size and disparate antennal sizes, and vice versa. However, we have now provided ANOVA statistical comparisons for Fig. 2D, Fig. 3D and SFig. 2F in order to allow more than just pairwise comparisons. Thank you for this suggestion, and we hope this addresses your concerns about more global comparisons.

3. The authors use simple behavioral assays to support the idea that there is a correlation between sensory specialization at the neural and behavioral level. However, one assumption is that quantitative differences in eye size or antennal sensilla translate to better olfactory tracking or color discrimination. This of course is not necessarily true--the species-specific expansion of ommatidia may not be used to extend color vision but for higher visual acuity or motion detection. We are not able to test at this time the specific behavioral ramifications of ommatidia increases, such as those mentioned, but would like to see this tested in the future (similar to a new paper by Ramaekers et al. 2018, bioRxiv, where they test visual acuity between two *Drosophila* species using stripe-width for optomotor response). We have added more to the discussion about the possible changes larger eyes can represent, such as color vision, visual acuity or motion detection, and mention that additional work is required to assess what is different in larger-eyed species (for example, if ommatidia increases are uniform, or if specific types are increased). Likewise, the choice of odor stimuli in the trap assay appears unrelated to the ecology of the species used and may not reflect quantitative

differences in sensilla. For example *suzukii* and *melanogaster* are likely to be attracted to qualitatively different olfactory cues (fresh vs. rotting fruit) and so would be unlikely to equivalently respond to the same set of stimuli. All *Drosophila* species appear to be rather strongly attracted to vinegar, hence the common name vinegar flies, which we show in attraction assays as identical for all tested species in Fig. 4E (when odor and color are presented together), but we also tested other fruit odors, with similar result between these species (SFig. 4A). Thus we do not feel there is a difference in these odors for host attraction (research has shown that *D.suzukii* has the greatest shift in chemosensation as related to oviposition preference, but we do not test oviposition here, and *D. suzukii* has been shown to be equally attracted to fermenting fruit as *D.melanogaster* for feeding preference; Keesey et al. 2015, J Chem Ecol; Karageorgi et al. 2017, Current Biology). Differences in behavior could therefore reflect differences in odor tuning not differences in sensilla number as proposed. Indeed, the expansion of olfactory biased species seems to reflect larger numbers of trichoid sensilla, which in *melanogaster*, are thought to be tuned to pheromone, not the food odors used in their assay. Olfactory expansion could therefore be purely related to differences in chemical communication between species.

(Copied from response to Reviewer #1) For this study we tested 3 main food odor sets for behavior (Fig. 4E-G, SFig. 4A), and did not find a significant difference between these odors, which we believe activate basiconic sensilla. Moreover, we see a consistent response when the visual and olfactory cues are presented together for all tested species (Fig. 4E), as in this case, *Drosophila* are all attracted to vinegar, hence the common name of "vinegar flies" for this genera appears to be accurate. A subsequent and unpublished study from our lab will outline specifically the trichoid and trichoid-related odors from a similar set of 60+ species of *Drosophila*, which we expect to see published in the coming 6-12 months, thus we have left out trichoid-specific odors and behavior from the present study.

We would also draw your attention again to SFig. 2E,F where you can observe that basiconic, coeloconic as well as trichoid numbers appear to differ between the measured species. Additional work would be needed to test changes in sensillum type according to host, habitat or mate choice. Additional work is also required to assess if changes (for example in basiconic sensilla types) are uniform, or if only specific ORNs are overexpressed during these changes in sensilla numbers between species. Genomic data suggests duplications of several receptors for example in *D. suzukii*, where one possibility is that these duplications mirror increases in sensillum types; however, again, additional work will need to be done to confirm these hypotheses.

4. The idea that wing patterns are prevalent in visual species and related to mate preferences seems an over-interpretation. Many species in rely on vision but do not have patterned wings. With the exception of *suzukii*, few of the visually biased species have sexually dimorphic patterned wings making their role in courtship less clear.

Thank you. We have subsequently revisited the data with new statistical tests for both wing pigmentation and light/dark courtship data, and we have added this statement to the results: "There was a significant correlation between female wing pigmentation and EF ratio after phylogenetic correction ($p = 0.04291$) (Supplemental Figure 3 H,I). In addition, there was a significant correlation between male pigmentation and EF ratio after phylogenetic correction ($p = 0.02557$); therefore, because there was a correlation between wing pigmentation and EF ratio when we include the phylogenetic correction, the correlation between these two traits has no phylogenetic signal (i.e.

the covariance of the residuals for the EF ratio and wing pigmentation regression do not follow phylogenetic signal). From the analyses of the light/dark courtship data in regard to EF ratio, we found these traits were strongly correlated both before phylogenetic correction ($p < 0.0001$) as well as after the correction based on relatedness of the species ($p = 2.406e-07$); therefore, we conclude from the available light/dark courtship data that our analyses supports the conclusion that trait correlation is not fully following a phylogenetic model - in this case a Brownian motion model of phylogenetic relatedness. (Supplemental Figure 3 H,I)."

Moreover, there are extensive studies on wing spots and courtship, where it is rare for females to wing fan or display their wings or participate in courtship dancing (even in species where females have pigmentation), thus we still hypothesis that wing pigmentation is more important for males in regard to courtship display and perhaps ultimately, species recognition. We have added all these points to the discussion about the possible ecological value of eye size in regard to, for example, mating and courtship, but more behavioral work is encouraged in the future to address these specific evolutionary pressures, such as visual courtship or sexual selection, especially in specific species. Again, thank you for your comments and we hope these new analyses add some weight to our ideas.

5. The authors suggest that the tradeoff between vision and olfaction is necessitated by the sharing of a single developmental structure—the imaginal disc. This argument arises from the existence of a mutant (Lobe) where eye progenitor tissue is lost while antennal tissue is expanded. However, not discussed is the observation that other mutations that showed reduced eye size but did not affect trichoid number (gl, gla) and mutations affecting trichoid number did not show a corresponding increase in ommatidia number (DII), leaving doubt as to whether affect of L mutations on trichoid number are truly reflective of developmental trade offs between eye-antennal structures as opposed to a more specific phenotype relating to this mutation. Given the wealth of tools available in melanogaster that allow for the expansion or restriction of growth in specific cells, the test of their model using these mutations feels unconvincing. Furthermore, the author's model of a developmental trade off. If the model is that the shared origin of the eye and antennae creates a competition for the pool of undifferentiated progenitor cells each organ has to recruit from then wouldn't the other structures arising from the eye-antennal disc (i.e. the maxillary palps, ocelli, head capsule) show the same inverse relationship? Finally, it would be good to clarify the difference between atrophy of a sensory system that is not used and the inversely coordinated tradeoff proposed here. For example, if we assume that a species relies more heavily in vision, the size of its eyes may be increased while they rely less on other modalities and are therefore smaller. However, this does not necessarily arise from a constraint.

In the time since initial submission, a second paper has appeared as a pre-print on bioRxiv, which much more clearly ties the developmental genetics to the observed tradeoffs between eye and antenna (Ramaekers et al., "Altering the temporal regulation of one transcription factor drives sensory trade-off", 2018, <https://www.biorxiv.org/content/biorxiv/early/2018/06/15/348375.full.pdf>).

The mentioned paper provides very convincing evidence through CRISPR-cas9 mutants and other molecular genetic tools afforded by the *Drosophila* model to provide strong support for the eye-antennal imaginal disc being responsible for a sensory tradeoff between these two structures in the adult. While the authors of this preprint only examine two species, *D.melanogaster* and *D.pseudoobscura*, we feel the evidence from our 62 species strongly match their conclusions, and we feel these two papers enhance and strengthen each other, as our manuscript provides far more

ecological, evolutionary, behavioral and morphological (internal and external) evidence for this sensory tradeoff, while their new preprint provides greater depth for the genetic mechanism and developmental constraints that underlie this tradeoff.

Due to the appearance of this preprint, we have elected not to pursue additional experiments during the revision process in regard to developmental genetics, as it was serendipitous that these researchers have now provided extensive support for the imaginal disc tradeoff hypothesis.

We have added additional discussion to account for your points concerning sensory atrophy, as well as hypothesize potential differences should ocelli, palps or other body regions be measured in addition to the eye and antenna (and we provide raw data images to begin to address this in the future). You also raise the point that eye and antenna potentially have different flexibility or plasticity, and that this may be different in each species (or perhaps a certain minimum or maximum value that directs or limits variation during this tradeoff), we have also added this idea to the discussion, thank you. We hope this manuscript is a first step in addressing more specific evolutionary questions, especially as most variation in EF ratio appears to be within species subgroups rather than between subgroups, suggesting perhaps that there are evolutionary pressures beyond host, habitat, or ecology, and that some pressures arise from competition or species identity.

Reviewer #3 (Remarks to the Author):

My main concern is there is no clear description of how the data was analysed. Without understanding how the authors analysed the data it is difficult to comment in depth on their results. Of primary concern is whether the authors analysed their data within a phylogenetic framework? It is clear they have a phylogeny but were their correlations performed accounting for phylogeny? There are also methods such as ancestral trait reconstruction which may help the authors consider their results within an evolutionary framework.

Thank you. We have provided a more detailed description of the generation of the new molecular phylogeny, as well as addressed the main trait (Eye-Funiculus ratio) in accordance with a phylogenetic correction. In order to provide further clarity and transparency, we have tried to expand the methods section, as well as include the raw data used for each analysis, including R-scripts for statistical values (see DOI link for raw data availability).

Given the authors are embarking on an evolutionary and ecological framework the authors need to be upfront about the limitations around the use of stock centre flies.

We have briefly included this point in the discussion.

["It is also important to mention that there are some limitations in our extrapolation to true wildtype insects due to the usage of stock center or laboratory flies, but we anticipate that our findings will extend to natural populations as well."]

The authors are claiming a trade-off between eye size and funiculus surface area they need to tread a little carefully here. Whether the observed correlational between eye size and funiculus surface area represents a functional trade-off or selection, cannot be distinguished across the species. It could possibly be that selection rarely operates in the direction of big eyes big antennae and these structures are costly to maintain such that relaxed selection would result in a reduction in size/use of

these structures. The mutant work, only in *D. melanogaster*, lends further support for a functional trade-off but I found this section to be poorly describe and thus difficult to interpret.

While we cannot directly state the pressures leading to changes in eye and antennal sizes across the 62 species, we do provide additional evidence through the new phylogenetic analyses that variation in EF ratio is not predicted by ecology, environment, or the relatedness of the tested species. Moreover, before we could address additional mutant work concerning the imaginal disc, a second manuscript has appeared as a preprint in bioRxiv, which greatly enhances the support for our conclusion of a developmental tradeoff or constraint between these two sensory structures.

(Ramaekers et al., "Altering the temporal regulation of one transcription factor drives sensory trade-off", 2018, <https://www.biorxiv.org/content/biorxiv/early/2018/06/15/348375.full.pdf>).

While the authors of this study only focus on two species (*D. melanogaster* and *D. pseudoobscura*), they elegantly show that a specific amino acid change alters not the function of a gene but the timing or onset, which in turn accounts for differences in the division between eye and antenna during development, where they also conclude that this results in a tradeoff between the two sensory structures due to the sharing of a single developmental structure.

It is clear the authors have done a lot of work but there were an overwhelming number of figures and I found it difficult to understand how it all came together.

We have tried to streamline the figures and the text to provide a clearer and more logical pathway through the immense amount of data provided. That being said, we hope the data provided, including as well now the raw images through the supplemental files, will provide a foundation of knowledge about many novel non-*melanogaster* species, and pave the way for new research to develop to address big picture ecological and evolutionary topics using this genus of flies as a model.

We are currently uploading about 150GB of raw data onto a new platform offered through the Max Planck Society for data sharing. A DOI link to the raw data files will be included with the resubmission, which should be active before the article would appear online.

Line 67 This is a vague statement, it is unclear how an inter-specific comparison will shed light on the field of neurotheology.

Thank you. This has been adjusted in the text.

[Consequently, as the field of neuroethology moves in the direction of understanding and incorporating the roles of multimodal signals for behavioral decision-making (i.e. visual, olfactory, gustatory, mechanosensory and auditory cues), similarly, neuroethology is also beginning to examine a multitude of closely related animal species for evolutionary comparisons of morphology, behavior and adaptation 13–15, which can help identify the selective pressures that drive these changes in sensory systems and neural development.]

Roughly how long has each species been maintained under laboratory conditions? Can the authors please add this to their supplementary table.

To avoid adding additional data to an already large amount of figures and subfigures, we provided specific stock numbers for each species. These stock identifiers provide greater details about time of

laboratory establishment, location of field collection, as well as information about reference specimens. You can read more detail about each species through the stock center website: (<http://blogs.cornell.edu/drosophila/orders/>) by following the link for "inventory list".

We also mention in the discussion that there are limitations in the use of laboratory or "stock" specimens that are maintained for multiple generations under laboratory conditions.

Line 130 I assume the authors mean head size and funiculus are more strongly correlated than body size and funiculus size.

Yes, thank you. Clarification provided in the text. As you point out, we meant that head size, as opposed to body size, was better correlated with eye and antenna values.

Line 275 There is no mention of candidate genes anywhere else in the manuscript. I am unsure how this section fits with their work as there is no clear methods for what they did.

Clarification has been added to the main text. Here we addressed candidate mutants within *D.melanogaster* (from publically available or existing stocks) for their effect on eye and antennal sizes (e.g. ommatidia and sensilla numbers). During the revision process, while we were pursuing more specific developmental genes responsible for our observed tradeoff, a second paper has appeared that more clearly identifies and tests a specific mutation that explains the observed Eye-Funiculus ratio. We cite this paper from bioRxiv in many of the other reviewer comments (where we hope our answers are shared with all reviewers), as this new paper provides more in-depth molecular genetics than we could generate within the time constraints allotted for resubmission. This new paper was rather serendipitous, in that we feel this other paper provides greater detail and methodology to explain the genetics of this tradeoff, while our own current manuscript provides 60 more example species, as well as brain reconstructions, and more ecological, evolutionary and behavioral consequences of any changes to these two sensory systems across the entire genus of *Drosophila*. Thus we are happy that a second group has found a similar phenomenon and that their work nicely compliments our own and vice versa.

Line 316 The authors have the data to ask some interesting questions about the origins and evolution of these structures across the *Drosophila* phylogeny but fail to shed any real light on evolutionary process across the phylogeny.

We hope that our manuscript, while not answering all possible evolutionary questions, provides instead a foothold to address these types of large-scale evolutionary questions through providing the first dataset to encompass such a large array of species within a recognized model genus, as well as provide the rationale for expanding this model from a singular species into an entire genus in order to address these types of interesting evolutionary questions. We feel this paper is a first step forward in an ongoing process towards pushing a new frontier in ecological and evolutionary research by allowing the most advanced molecular and genetic tools to be applied directly to these new species via their relation to *D.melanogaster*, which is a long-standing genetic model organism. Moreover, I hope this manuscript provides data and ideas that promote additional research in this field. We agree, for example, that more work needs to be done to understand the pressures that lead to this tradeoff, but we feel strongly that our current manuscript strongly identifies that this tradeoff exists, and that this area of study requires additional research in the future.

Line 321 The authors introduce new data in the discussion? There could also be a power issue here?

Here we provided already published data to support our conclusions pertaining to an existing pseudo-gene hypothesis (where we argue against the idea that olfactory pseudogenes correlate with visual enhancement). Sadly, for most *Drosophila* species we know more in regard to their genetics than their ecology, evolution, or developmental biology. However, we still feel our manuscript provides a great start to this push for more studies about the vast array of non-model species within this genus of fly, and we hope others find novel directions to pursue from the data provided.

Methods

Did the authors control for density which would have clear implications for body size?

Rearing was controlled for density, yes, with between 20 and 25 females per vial. The within species variance in measurement of Eye-Funiculus ratio is quite reasonable and consistent, as can be seen in Fig. 4C for all 62 species.

The methods section lacked any clear explanation for the purpose of their assays, making it difficult to determine why they did what they did. Simple sentences at the start such as To examine..... would greatly improve the readability and understanding/justification of their methods and conclusions. For example I found the methods for the behavioural assays confusing with no clear defined purpose for these assays. I am unsure to what end the authors were aiming to achieve with these assays.

Thank you, we have tried to rewrite sections of the methods for better clarity and readability, though we are heavily restricted based on word limitations. As for the behavioral tests, we sought to address possible implications for shifts in either eye or antenna size, including ramifications for both host-seeking behavior (Fig. 4D-G) and courtship (Fig. 4B); however, a more extensive study is underway for the latter, which we could not overlap objectives with, thus we have not sought to include additional data concerning pheromone production, detection or behavior.

How did the authors analyse their data? I cannot seem to find this either in their methods or their supplementary materials.

We address analysis for external morphological measurements, or internal morphometrics such as AL and OL. However, to assist in clarity, we have written an additional methods section on statistical analyses. We also provide all raw data, and where possible, R-scripts as well, in order for readers to recreate or reanalyze the raw data, or to use our data in new meta-analyses. We hope this provides a reasonable solution to these concerns. Thank you again for your comments and suggestions.

Reviewer #4 (Remarks to the Author):

Keese et al present an analysis of investment in visual and olfactory sensory structures across a large number of *Drosophila*. Their results imply a trade-off between sensory modalities, and the authors suggest that this is evident in the external anatomy, neuroanatomy and neural development of these species.

There is a lot to admire about this paper. It contains a huge amount of work, combining different neuroethological approaches in a broad comparative context. It is a hugely ambitious study that, off the top of my head, is unique among the invertebrate literature, and is exactly the kind of comparative neurobiology that I think is needed in an era dominated by model organisms. I was wholly ready to absolutely love the paper. Unfortunately, I don't – yet! I do think there is an excellent paper here, but the current version has a number of issues pertaining to the data analysis and

interpretation. As a result, I think the paper doesn't quite do the idea or data justice. However, these issues should be relatively straightforward to address and I anticipate that the revised manuscript can be much improved.

I'll try to explain the general issues I have, then provide some more minor, specific comments. The main issues are:

- 1) a lack of phylogenetic correction
- 2) use of ratios
- 3) inference of trade-offs

1) Phylogenetic correction: the introduction nicely sets up the idea of expanding the neuroethology approach across a phylogenetic scale to investigate evolutionary questions. However, evolutionary biologists recognise that species are non-independent due to the hierarchical nature of their phylogenetic relatedness. As such, interspecific data cannot be analysed using standard statistical methods. Phylogenetic non-independence is important as it can effect not only the significance of any regression analysis, for example, but also the slope and residual variance. Statistical approaches that remove, or correct, for the effects of phylogenetic non-independence are needed. There are several ways of doing so now, implemented in quite accessible packages such as BayesTraits, Phylotools, APE. Essentially all the statistical analyses in this paper should be re-done using these methods. This would include the regressions in the figures.

Thank you for your extensive comments and suggestions. While this area of phylogenetic correction is new for our team, we have now sought to account for and reanalyze the data with these factors in mind. Using the R statistical program in association with Phytools, Ape, and Caper, we have tested a newly created molecular phylogeny (now included in supplemental files) using two separate phylogenetic tests, including Blomberg K and Pagel's lambda ($K = 0.4783$; $p = 0.041$; $\lambda = 7.102e-05$; $p = 1$), which both agree, that our main trait (the Eye-Funiculus ratio) is not supported by the phylogeny. Moreover, we present the trait as it fits into the reconstructed phylogeny (Fig. 4 A), using both size and color to indicate the value for each species. Here it is apparent (and matches the Blomberg K-value of less than 1) that variation is high within subgroups, more so perhaps than between subgroups. While we could only locate and assess nuclear and mitochondrial genes for 59 of the 62 species (e.g. we are missing 2 subspecies of *D.mojavensis* as well as another individual species, *D.montium*), we feel this reanalysis is adequate to extrapolate towards the complete list of 62 species where we have additional morphometrics elsewhere in the manuscript. Again, we do not find a phylogenetic correlation with our EF ratio, suggesting that the phylogeny does not explain the trait variation we observe.

As we do not find a consistently statistically significant association between the phylogeny and the Eye-Funiculus trait using two different tests (Blomberg and Pagel), we feel it is reasonable for comparing species in the regressions without additional correction. We do additional phylogenetic testing when analyzing both wing pigmentation and light/dark courtship data against our EF ratio trait. However, we also provide all raw data for transparency and for additional tests to be conducted in the future. This includes R-scripts and other raw data files with the online version of the paper.

In addition, as this manuscript is just the beginning of this area of research, we hope the tests that appear in the current manuscript are sufficient to move the field of study forward, and admit that our expertise is not adequate to test all evolutionary scenarios. We hope researchers are able to use our

data in the future as a foothold, and are able to successfully test additional hypotheses using our data, or to augment their own data with that which we include. For example for new metrics like palps and ocelli measurements, or to reanalyze brain reconstructions to address differences in other regions, such as the mushroom body and lateral horn.

2) Use of ratios: Throughout the paper the authors use a ratio of two traits for comparisons of investment, and to correct for allometric effects caused by differences in body size or brain size, for example. This is very common in the comparative neurobiology literature but, in my opinion, is quite flawed. Using a ratio assumes your two variables scale isometrically with a slope of 1. This assumption does not hold in the majority of biological cases (see, for example, Voje 2016, *Am Nat* 187(1)). Without wanting to be patronising, I will try to explain why I think this is important. If your traits do not scale isometrically the slope of the regression will be significantly greater or less than one. This slope is interesting in itself as it can be used to infer some predictions about the constraints involved in the relationship between the two traits, but it also affects the inference of proportions. If two traits scale hypo-allometrically, with a slope of <1 , assuming isometric scaling will make it look like the Y variable in larger individuals/species are disproportionately small compared to smaller individuals/species. The converse is true for hyper-allometry. As such, ratios can produce misleading measures of relative size. Of course, the authors may argue ratios capture the percentage investment level from a mixed energy budget. However, this assumes the traits are free to vary in that way. If they scale non-isometrically, they may not be able to. The proportional size may therefore not reflect an investment trade-off, but may instead be determined by some functional scaling effect that is not related to energetics.

I've written a couple of small things about these effects in other context elsewhere; in the hope that these may be helpful, see: Montgomery 2013 *Brains Behav Evol* 82 (3); Montgomery & Mank 2016 *Mol Ecol* 25 (20). I would replace all the analyses with ratios with multiple regressions (in a phylogenetic context). Where the authors want to plot something like a ratio I'd perform a phylogenetic regression and plot the residuals. Of course, the alternative would be to test for isometry in the data – if your traits do scale isometrically the use of ratios is probably justified. However, even here there are problems - the other thing to point out is that when the authors do not have an independent variable (e.g. central brain size), so are comparing OL/AL or imaginal disc compound eye/antenna for example, there inevitably has to be a negative relationship as if one trait gets bigger the other one will 'appear' smaller, even if it has in fact stayed the same size. So, in sum I would not use ratios to infer trade-offs.

Currently we are not able to refute this argument that ratios, while the currently established norm across this field of study, are not optimal for addressing all hypotheses related to evolutionary investment. We fully support additional testing using the raw data we provide, but feel strongly that our conclusions are ultimately correct and well-supported by the types of analyses and measurements for the traits we examine.

3) Inference of trade-offs: I'm not 100% on board with the conclusion that there is a trade-off between the sensory modalities for a couple of reasons. One is statistical, I'd like to confirm the negative relationship correcting for the issues above, but I imagine it will hold. The second is ecological. If I have followed the authors argument correctly I think they are saying the trade-off reflects competing demands of energetic investment (i.e. the flies have to 'choose' between visual investment and olfactory investment). However, later the argument seems to morph into the trade-off existing due to negative genetic covariance (as suggested by the development/mutant lines). I

don't think either is necessarily convincing. First, the alternative explanation is simply that these species evolved in different habitats and these habitats select for differential investment in each sensory modality. In some habitats visual information is more reliable – so selection favours increases in visual structures – but olfactory information is less reliable – so selection favours decreases in olfactory structures. In other habitats the opposite is true. Using the phylogeny and trait values in Fig. 4A, we argue that variance in Eye-Funiculus ratio does not match ecological niche or habitat selection pressures, nor does it match phylogenetic relatedness. For example, cactophilic flies show similar variation in Eye-Funiculus ratio as within fruit-feeding guilds, or across sap-feeding species. While we cannot address all possible sources of evolutionary pressure (partially as you point out, because there is a paucity of ecological information available for the majority of the examined species), we still feel the data we provide captures that a tradeoff occurs between the eye and antenna, which is not supported by phylogenetic relationships or ecology. Future work is required to assess the causality of the variation in these sensory systems. However, I would once again point to the recent preprint (Ramaekers et al, bioRxiv, 2018), which also reaches the same conclusions we do in our manuscript, namely that a developmental constraint and subsequent tradeoff is encapsulated by the sharing of a common structure, the eye-antennal imaginal disc, during larval development. We concede that additional work is needed to determine which selection pressures drive this tradeoff, but hint in the discussion that we believe it is likely competition or species recognition.

In the discussion the authors use variation across Lepidoptera as another example of trade-offs between the senses. I actually think this better illustrates the role of ecology. Yes, OL and AL size are negatively correlated across Leps, but when you look at the detail the variation is much more subtle. For example, there are diurnal butterflies that invest in AL size to a similar extent as some nocturnal moths, while still investing in the big OLs typical of butterflies (see, for example Montgomery & Ott, 2015 J Comp Neural 523:869-891). So in my view, selection seems to be able to bring about increases in both sensory modalities, if it wants to (this is quite an adaptationalist interpretation, admittedly, but here the trade-off is about utility not development or energetics).

The developmental work presented in the manuscript does not convince me that this could not also be the case here.

We agree, similar to your examples from Lepidopterans, that ecology and host choice do not seem to explain the variation in *Drosophila*. Until additional work is done to address behavioral and natural or fieldwork-related decisions for more species within this genus, we can only speculate as to the causes or pressures that predict or govern these shifts between the eye and antenna. There simply is not enough ecological data available for the majority of the species we examine, as again, we know more about the genome than we do about the host choice or mating preference of the species. One scenario we still feel strongly about is that character displacement might be a factor driving the species to be as different as possible from close relatives, either to avoid competition for host materials, or to avoid unsuccessful, or infertile mating between close relatives. This has been well-documented in Cerambycid beetles, where the insects use the timing of pheromone release (either daily or annually via emergence patterns) to help separate closely-related species that use similar pheromone blends (where cross species mating leads to infertile offspring or mistaken courtship and lost energy)

(Mitchell et al, J Chem Ecol, 2015: <https://link.springer.com/article/10.1007%2Fs10886-015-0571-0>)

First, all the imaginal disc work shows is that development mirrors the final phenotype – surely this is to be expected? It doesn't really show that two areas of the imaginal disc can't evolve independently.

Second, the mutant line work does provide a mechanism for linking the relative size of the two traits, but obviously does not rule out the possibility of there being other mechanisms that could permit independent evolution. To rule out an ecological explanation the authors would either have to show that there is no genetic variation in these species permitting independent evolution (very hard to do but you could take a quantitative genetics approach) or to incorporate ecological data in the analyses to see if this predicts the interspecific variation (perhaps also infeasible due to lack of data). Eco-Evo-Devo is a relative young field, and while *Drosophila melanogaster* offers a great array of molecular/genetic tools to address the imaginal disc, these tools are not yet as readily available for most non-*melanogaster* species. However, the recent advance in targeted gene manipulation through CRISPR has greatly assisted in addressing evolutionary shifts in specific molecular functions. Over the last few years, a CRISPR framework has been developed for several species within the genus, including *D.suzukii* (Karageorgi et al, Current Biology, 2017), *D.simulans* (Seeholzer et al, Nature, 2018), *D.subobscura* (Tanaka et al, Journal of Neuroscience, 2017) and *D.pseudoobscura* (Ramaekers et al., 2018), with more species in production, for example, *D.sechellia* (unpublished). It is again our hope, that the current manuscript will provide a starting place for ecological, behavioral, evolutionary and developmental questions and hypotheses that can be tested in the future using advanced molecular tools which have been developed previously for *D.melanogaster*. Thus our other goal for the current manuscript is to push this insect model from a singular species into an entire genus (a genus which spans thousands of species across an incredible array of hosts and habitats) and we hope this manuscript is the first of many large-scale, big-picture assessments of evolutionary principles in order to predict or determine variation in animal species both big and small.

So perhaps for now the authors can just discuss the alternative explanations in addition to their favoured interpretation. One analysis that may be possible to add would focus on intraspecific variation. If a genetic/developmental constrain is responsible for the trade-off you expect it to operate within individuals as well as between species. If the authors have large enough intra-specific datasets they could test whether or not the negative relationships are apparent within species, which may bolster their argument. If they are not consistently negative correlated, I'd probably interpret this as evidence against the current interpretation.

Thank you for this idea about intraspecific variation. Currently we do not have a robust enough dataset to test this idea adequately, though I think it would be great to address this in the future, as some species might show more plasticity than others in regard to either eye or antennal variation. Future work could examine things like population density or diet to see if there are distinct intraspecific changes in response to these factors.

More specific points (many of which relate to the above):

Line 66: "the study of a single species will also often be insufficient to address... evolutionary questions" – I would say never! Unless you're studying inter-population differences and micro-evolutionary process. From a comparative perspective, one species is $n = 1$ so zero statistical/inferential power.

Agreed. We have corrected the writing to emphasize your point here.

Line 129-133: here multiple regressions may be useful to test whether body size has any effect (also on line 140). I wasn't sure what 'zone 1' and 'zone 2' mean here (biologically). If the point is just to show there is variation you could estimate the regression parameters and plot the residual variance.

As the variance is much reduced in panel C I would guess a lot of the effects relate to variation in selection shaping body size, rather than eye surface area.

We have removed the zones, and agree the data better represent variation in body size, which we now mention in the text.

Line 142: its not clear what you mean by species pairs – I tried to locate these 6 species on the phylogeny (which took a while - maybe you could label them as you have done for other comparisons) and (I think) they do not form phylogenetically independent pairs (of red/blue datapoints), which is what I assumed you meant. As such your regression will be affected by phylogenetic non-independence here as well. As the text is written now, *D. americana* also appears in both groupings – which must be an error since you say ‘vice-versa’

Originally, we selected 4 pairs for deeper and more time-consuming measurements and analyses, thus we also originally arranged the statistical metrics in pair-wise fashion. However, as you and other reviewers point out, this limits the global comparisons, and as such, we have instead provided the data in a format that is more conducive to comparison across all species (and used ANOVA for this reanalyses). We did in fact, use *D. americana* for two separate comparisons (see Fig. 2A), first across similar funiculus size but disparate eye size (with *D. busckii*), and then in a second comparison across similar eye size but disparate funiculus size (with *D. funebris*). However, again, we have provided these comparisons and more through the removal of the pairwise comparisons. Please see Fig. 2D for more complete comparisons among the focal six species, and again in Fig. 3D for brain-related morphometrics.

Line 162-163: what about ommatidium size? If surface area and number both vary you could calculate estimates for this. I think this would be interesting. If facet area is larger it may imply adaptations for photon capture, whereas increases in facet density may suggest selection for greater acuity (maybe?). You might also predict this would affect neuropil differently. For example, increasing facet number without increasing facet size should increase lamina/medulla volume. Whereas increasing facet size without increasing facet number might not as there would be the same number of cartridges.

Due to text length limitations, we were not able to directly address all data extrapolations or hypotheses, but we encourage the use of the data provided to examine these and other interesting ideas! We especially like the idea of estimating ommatidia number from measures of eye surface area, but we are not able to specifically examine this in the current results or discussion sections.

Line 168-192: Why was the lamina not measured? Is it damaged in the dissections?

We were not able to be confident in measurements due to dissection procedures.

Line 176: when using central brain volume as a control, do you subtract AL volume?

Correct, hemisphere values first subtract AL volume. We have added this detail to the methods section for additional clarity.

Line 184: it looks like ME, LOB and LOP may vary somewhat independently of total OL. Is this not of interest? Total OL size will also be dominated by the effects of ME variation, so if LOB and LOP are varying independently this could be masked to some extent.

We agree with your assessment of Fig. 3E, but again, due to word limitations we could not specifically address this observation. However, we again encourage additional testing using the

provided raw data to test these great ideas!

Line 189: “we note that the central brain hemisphere as part of the whole brain was consistent in size across all tested species” – I’m not really sure what panel E shows. If the values are percentages they seem very small? Also, if this was true then there would be no effect of correcting for central brain size as the denominator would shift numerators in a consistent way. So are you really just seeing effects of absolute size of the neuropil? If so, do these vary negatively? If not, is there any real trade-off?

Panel E in Fig.3 is a percentage of central brain or hemisphere volume (after subtracting AL) that each section of the OL encompasses per species. We consistently find larger eyed (or larger Eye-Funiculus) species to have increased relative percentages of these three sections of the OL, while a trend towards diminished AL percentages. The absolute neuropil size was more similar for small Eye-Funiculus species and again for large Eye-Funiculus species (Supplemental Fig. 3D,G); however, due to the massive size of *D.pseudotalamancana*, this species was a clear outlier in absolute sizes of the brain sections. Again, we believe correcting for size of the hemisphere provides an accurate way to compare AL and OL of these species despite variation in absolute size of the neuropil.

Panel E in Supplemental Fig. 3 is percentage of the central brain or hemisphere in relation to total brain size (e.g. OL, AL and central hemisphere combined). This value appears to suggest that hemisphere size is consistently related to sizes of sensory inputs, such as eye and antenna. We provide the raw scans, and feel additional analyses could be done to compare other brain regions, such as lateral horn or mushroom body for example; however, these types of comparisons were beyond the initial scope of our study.

Line 223: “based on our arrangement” – I don’t get why the authors don’t do any analyses here. An arrangement of data doesn’t provide any evidence in support of their hypothesis. You could easily do a (phylogenetic) multiple regression with eye size and/or funiculus size with binary variables for wing pigmentation and/or courtship to test if these ecological traits explain variation in sensory investment. As it is, its just descriptive (fine, but not convincing).

We believe this comment is in reference to line 233. During the revision stage, we have now built a molecular phylogeny, and provided statistical tests to address wing pigmentation and light/dark courtship in association with the Eye-Funiculus ratio trait. These types of analyses were new for our team, but we have brought on additional authors to assist in these types of statistics. We found that male pigmentation was significantly correlation with EF ratio, though not after phylogenetic correction. In addition, we find statistically significant support for light/dark courtship in relation to this EF ratio trait, where larger EF ratio matches reliance on light for successful courtship. As this type of manuscript and dataset is new for our team, we took longer to figure out how to accurately analyze these types of questions, thus thank you again for your time and assistance in providing suggestions for how to examine these questions in our reanalyses of the dataset!

Line 256: “species that differ drastically in absolute size” – I think this is a key point in absolute terms both traits are increasing in size/number. Using a ratio doesn’t really correct for these allometric effects. As the analyses are presented I think it remains possible both are increasing independently, but to differing degrees.

While we cannot refute the possibility that eye and antenna vary at something other than a 1:1 ratio in this tradeoff, we continue to assert that we conclusively demonstrate the tradeoff exists between

the two traits and that this is supported by our data and analyses, as well as supported by the recent pre-print in bioRxiv (Ramaekers et al., 2018). As per a similar response to a previous reviewer comment, there is a chance that the eye or antenna within the imaginal disc differ in their flexibility or plasticity between species, though additional work would need to be done to address this in more detail. But again, our manuscript provides the first evidence for such a tradeoff to exist in *Drosophila*, and subsequent papers can focus on more specific mechanistic, species-specific variation, and address other questions about the circumstances or causes leading to this tradeoff across the genus, such as species-specific plasticity or limitations for either eye or antennal shifts.

Line 264: “structures would essentially be competing for the same resources within a single disc” – only if there are no developmental mechanisms that can promote localised proliferation/growth. Future work will need to confirm all species, but at least for *D.melanogaster* and *D.pseudoobscura*, the developmental mechanism for the observed variation supports our conclusion for a sensory tradeoff between the eye and antenna based on competition for resources within a single disc (Ramaekers et al., bioRxiv, 2018).

Lines 316-331: I find this paragraph unconvincing, in particular the correlations with the number of sensory genes. I'd remove it as it doesn't show much anyway (and suffers from phylogenetic issues). Its unlikely this idea holds up – for example in Leps, OL size varies hugely between nocturnal/diurnal species, but most species have 65-70 OR genes.

We use this section to address existing literature that also examines the potential for a tradeoff between visual and olfactory sensory systems in other animals, such as primates and additional insect species. We have also augmented this section with new examples from *Drosophila* (Ramaekers et al, bioRxiv, 2018), and stickleback fish (Keagy et al, “Brain differences in ecologically differentiated sticklebacks, Current Zoology, 2018). We feel this section in the writing gives some background and context that this phenomenon has been proposed previously, just not in a model organism with the tools afforded by *Drosophila melanogaster* in order to adequately test it.

Another important point is that previous papers have usually focused on presence or absence of genes (for example, olfactory pseudogenes in primates); however, variation has also been shown to be of evolutionary importance in regard to expression of these genes, such as in the ab3 sensillum (Or22a) overexpression in *D.sechellia* which is associated with its specialization on morinda fruit (and the detection of methyl hexanoate and hexanoic acid), or the reduction in at1 sensillum (Or67d) expression in *D.suzukii* that is associated with this insects shift away from cVA (a male-specific pheromone) in behavioral attraction. Thus we agree, the number of different gene types is not necessarily sufficient or interesting to compare (in your example, between nocturnal and diurnal relatives), but expression levels of those genes might be, which we provide some measure or estimate of through ommatidia and sensilla counts for a number of novel *Drosophila* species. Future work could examine which ommatidia are expanded, for example, those related to visual acuity, motion detection, or perhaps color-specific sensitivity, or counter to that, perhaps ommatidia types are uniformly increased in larger-eyed species. A similar study of OR expression could be done along the antenna of novel species, as our data clearly show shifts in the number of basiconic, coeloconic and trichoid sensilla for many species, but we do not yet know if that is uniform or specific in regard to gene expression changes. As an example, our data support the work on *D.suzukii* (Dekker et al., Proceedings of the Royal Society B, 2015), which outline that there is a reduction in at1 sensilla and a reduction in volume of the associated glomerulus in the AL; however, our data go further in showing

that *D.suzukii* has huge increases in basiconic as well as coeloconic sensillum types, at least numerically. But we cannot at this time say whether those increases are biased towards a certain ORNs, as we do not have expression level data available, though this would be interesting to assess in the future. Again, while *D.suzukii* has been heavily studied in regard to the OR genes it has relative to *D.melanogaster*, no study has thus far compared expression levels, or functional data, which might ultimately prove to be a more important measure for evolutionary shifts in ecology or host choice across Drosophilidae, and not just receptor identity alone.

Lines 376-397: see comments above on Lep brains.

Addressed above, where we measure overall AL volume, but not glomerulus volume (the latter of which might be indicative of expression or the number of a given receptor expressed on the antenna).

Figure 1: are the branch lengths meaningful? Panel A would make a great cover image!

Thank you! In this revision we now present a newly built phylogeny with meaningful branch length in Fig. 1B, and also feel the 62 species frontal views would make a great cover! We have proposed or attempted to generate a new cover suggestion that is included in the supplemental figures for consideration by the journal. Just to repeat again, all accession numbers for gene sequences and all data required to build or examine the phylogeny are now included with the online version of this manuscript via a DOI supplied by data storage associated with the Max Planck Society.

Figure 3: can you add the sample size for each species (as in figure 2G). The pairing of species in panel D is also a bit random. For example, americana is more closely related to funebris and pseudo. than it is to busckii. Indeed the closest phylogenetic pairing (excluding melanogaster/suzukii) would be americana/pseudo.

We have added sample size to all figures directly, or within the figure legends. In addition, all raw data is included for additional dissection or novel analyses. Pairings have been removed in the analyses you point out, and instead, more global ANOVA comparisons have been introduced into the paper for more comparisons between other species.

Data: Perhaps I missed it but I didn't see any reference to data accessibility – can the raw data be made available?

Yes! We have been working tirelessly the last few months during the revision process to provide as much as data as possible, using a relatively new resource from the MPG, called "Edmond". We now have a website DOI which we are currently curating to include and organize all raw data from this manuscript (including images, sequences, measurements, reconstructions... in total almost 150 GB of raw data!)

I hope these comments are taken as constructively as they are intended! As I said, I think there is a great, exciting paper here once these issues are addresses. I'm happy for the authors to contact me if it would be helpful to discuss any of the issues further.

Thank you ever so much for the time and depth of your comments and suggestions. We also hope that the revised version of the manuscript will provide something more akin to your initial response to the manuscript, and we too would enjoy an ongoing dialog about this project and others.

Best,
Stephen Montgomery (shm37@cam.ac.uk)

Reviewers' Comments:

Reviewer #2:

Remarks to the Author:

In this work, Hansson and colleagues offer an impressively large and detailed analysis of peripheral sensory structures across a wide array of *Drosophila* species to support the compelling idea of a trade-off between visual and olfactory systems. I admire the ambition of this study to extend beyond model organisms and consider the constraints that bias the evolution of sensory systems. However, as is, it represents an intriguing but preliminary catalog of *Drosophila* sensory structures and neuroanatomy. While I appreciate the responses of the authors and their attempts to clarify and justify their conclusions in the revised manuscript, the fundamental concept that there is an inverse resource allocation to visual and olfactory structures that contribute to species-specific host-specialization or mating behaviors remains insufficiently substantiated.

For example, a central premise of the work is that the size of a sensory structure should linearly correlate with the behavioral relevance of that sensory modality for any particular species. The authors suggest that the density of neurons in these structures is relatively constant across species, implying that the increased size is accompanied by an increase in the number of sensory neurons. This fails to address the key point—quantitative differences in sensory neuron number do not necessarily correspond to qualitative differences in odor discrimination or sensitivity. One could argue that the diversity of olfactory receptors and not the number of sensory neurons is more relevant to a high functioning olfactory system. While the authors highlight the ecological framework of their work, their behavioral assays are too coarse to reveal any insight into how the selective pressures that may drive sensory specialization in different species that inhabit distinct niches.

Another concern is the lack of clarity of whether the tradeoff between these structures reflects a true developmental constraint. The authors cite a recent paper by Hassan and colleagues in bioRxiv to support their work, suggesting that they have not delved further into any developmental mechanism, "as our manuscript provides far more ecological, evolutionary, behavioral and morphological (internal and external) evidence for this sensory tradeoff". This bioRxiv preprint does indeed support a tradeoff in one pair of species—*melanogaster* and *pseudoobscura*—through a nice mechanistic analysis of a single transcription factor. However, remarkably these two species show a very similar EF ratio in the current study (Fig 4A), raising concerns about the sensitivity of their analysis and relevance to this current work.

Reviewer #3:

Remarks to the Author:

The authors have put in a lot of effort to address the concerns of the previous draft and have improved the manuscript, however a few concerns remain.

Although the authors have showed different lines of evidence to suggest a trade-off between the sensory organs, the strongest line of evidence still comes from the species correlations. While I think it is fine to propose a trade-off the authors should acknowledge that they can not confirm a trade-off with their current data and while the Ramaekers et al lends support to this theory this has only been shown in 2 species.

I don't think the authors adequately address reviewers 4's concern with the use of ratios. At the very

least they could show whether or not there is isometry in the traits of interest. And while I appreciate the authors have taken great means to address many of the statistical concerns of the past draft, a multiple regression approach should not be that difficult, particularly considering the authors are already performing phylogenetic regressions using the caper program.

Line 143 Did the authors account for multiple comparisons when performing correlations amongst traits

Figure 1b What are the colour codes on the phylogeny.

Figure 2 F and Figure 5 F How were these species chosen?

Figure 2B Why are some species blue and some red? There is not enough care in the preparation of the figure legends to clearly state the detail and purpose behind the detail in the figures. If the authors do not have enough space I suggest they simplify their figures to avoid confusion.

Line 151 it is unclear to me and from the methods what unique measurements were made on the 6 species. I can gather from

Figure 2 that these were the basiconic, coeloconic and trichoid, but then I am not sure why the authors present Figure 2B? they have sensillum and ommatidium counts for all 62 species?

Line 157 Which 6 species did they choose? Clearly not the ones in brackets because with mel and suzukii that would be 8 species? If the species they choose did not include the ones in brackets I suggest they remove these to avoid confusion.

Line 213 Your phylogeny comprised 59 of the 62 species?

Line 238 The wording around this is awkward, accounting for phylogeny did not alter the strength of the relationship between the two traits. This brings me to another point P values are relatively meaningless I am more interested in how much variation can be explained by the relationship, authors should report r^2 and slopes throughout the ms. Note phylogenetic controlled analyses should be the gold standard and reporting the r^2 , slope and P value from these analyses supersedes any standard regression analysis and is therefore all that needs to be presented. In general the authors can simplify this whole section to just say we accounted for phylogeny, there were no strong relationships and here was the result.

Line 297 How did the authors identify this multitude? Randomly I hope.

Line 363 How well are the olfactory genes and pseudogenes defined in the 14 Drosophila species? And are the authors tests robust enough to say pseudogenes are not playing a role? I think the authors need to be careful in their conclusions here.

Line 401 This sentence is oddly worded and thus confusing. I think the authors need to also be careful not to make too strong conclusions about their data. How robust is the courtship data across the 62 species? I think there are interesting patterns but the authors can't explain everything with their data, it is better to own the limitations than make bold and potentially incorrect statements.

Line 461 To bold I don't think you can confirm this. You have some good evidence in support of this theory.

Reviewer #4:

Remarks to the Author:

I have read the revised manuscript and authors' response to my comments. I still really like the paper, it has got a lot of interesting work in it. I don't agree with everything the authors say/do, but that doesn't mean they are wrong or that the paper isn't ready for publication. For the benefit of the editor I summarise the response to my main comments and add any remaining thoughts.

I made three major comments in my previous review, below I summarise the actions taken by the authors for each of these:

1) a lack of phylogenetic correction: the authors have produced a new phylogeny of the species in their dataset and used two methods to estimate the phylogenetic signal of their traits of interest. Both suggest phylogenetic signal is low for the key sensory traits, which is reasonable justification for not correcting for phylogeny in these cases. In other analyses the authors also present results from phylogenetic and non-phylogenetic analyses. They've done a good job on this critical issue.

2) use of ratios: the authors do not refute my suggestion that ratios can be misleading but keep this measure in their main analyses. As far as I can tell they do not add any analyses to explore this issue further. My view remains that this is a not ideal, but it is common practice.

3) inference of trade-offs: the authors provided an interesting response to my suggestion that the evidence for developmental trade-offs is not particularly strong, and that ecology/selection could produce similar patterns. But I'm not entirely convinced by the response; its true that the ecological data is insufficient to support this idea but I would suggest that the developmental data is as well. I don't agree the authors 'conclusively demonstrate' a developmental trade-off explains the interspecific data – they demonstrate a potential mechanism for such a trade-off which is highly intriguing, but they don't show that it restricts the action of selection. I've only skim read Ramaekers et al, but at first glance I'd probably make the same comment on that paper. I appreciate this is quite an adaptationalist argument but, to me, it seems reasonable to me to give both hypotheses (ecology/development) equal weighting and await new data in the future.

4) The minor comments are well addressed and the data should now be freely available once published, so maybe I will find the time to play with it myself!

In sum, comments 2 and 3 are left in a slightly unsatisfied state from my particular perspective. That said, there are mitigating circumstances as the data may not be easily available to follow these points up. Indeed, the authors seem to accept these limitations but rightly state that future work can build on their results/data to test the conclusions obtained from their data. As such, although I could continue to argue my case (ad nauseam...) the authors are entitled to state their own interpretation of the data and I don't think my disagreeing with it should impede publication of a generally very nice paper. I look forward to seeing it in print.

Minor comments

Line 156: in the 'pairs' are not monophyletic I wouldn't refer to them as pairs. I'd just say '6 species that include the range of variation seen across the genus' or something similar.

Line 235: The phylogenetic behavioural tests are interesting, but I'm not sure what "light or dark settings" (line 234) is exactly, light level?

Results: I think the text for 'Phylogenetic correction of eye to funiculus ratio' is maybe in the wrong position, its currently on page 7 but I would consider integrating it with the first results section on page 4/5.

General: standardise p-values to 3 decimal places for neatness?

Lines 243- 246: not sure this is phrased well, the data could have phylogenetic signal but still produce these results. I'd delete the sentence beginning "therefore, we..."

Lines 35-378: in primates there is good evidence this is not a 'trade-off' it's the result of independent selection pressures shaping olfaction and vision in different ecological niches, so I don't think this is analogous to your interpretation of the data. I'd still suggest deleting this paragraph, I think it is weak and doesn't add anything to the paper.

General: I'd use central brain throughout, and ditch 'hemisphere'. Its confusing to use two terms.

Fig 2F: is this regression Phylogenetically corrected? State in the figure legend

Fig 3: if these 6 species are not 3 monophyletic pairs (?) it is perhaps a bit misleading to present them like this?

Methods lines 953-962: I'm unsure what the p-values refer to, I think the authors did a test comparing an estimated phylogenetic signal to one where it was set to zero? If so, the Blomberg estimates still suggest there is some signal, so phylogeny should be accounted for.

Sample sizes: unless I'm being stupid, I don't see the sample sizes in all the figures/figure legends.

Line numbers refer to the “accepted” changes version of the written manuscript resubmission.

We have now created a digital library providing all the source data of the manuscript. We here provide login-information for your complete access:

Website: <http://doi.org/10.17617/3.1D>

Login: ikeesey@ice.mpg.de

Password: sourcedata2018

However, we are currently in process of making it open access (open copyright) for the public.

Reviewer #2 (Remarks to the Author):

In this work, Hansson and colleagues offer an impressively large and detailed analysis of peripheral sensory structures across a wide array of *Drosophila* species to support the compelling idea of a trade-off between visual and olfactory systems. I admire the ambition of this study to extend beyond model organisms and consider the constraints that bias the evolution of sensory systems. However, as is, it represents an intriguing but preliminary catalog of *Drosophila* sensory structures and neuroanatomy. While I appreciate the responses of the authors and their attempts to clarify and justify their conclusions in the revised manuscript, the fundamental concept that there is an inverse resource allocation to visual and olfactory structures that contribute to species-specific host-specialization or mating behaviors remains insufficiently substantiated.

Thank you for your comments. As we have taken a rather broad first approach (i.e. examining over 60 species), we consequently suffered from heavy limitations concerning our ability to run tests of all possible behavioral combinations, or tests that included the behaviors of every species. However, we feel that trap assays are the industry standard for examining host preference or attraction in *Drosophila* and we feel we have provided a sufficiently-replicated assessment of behavior across several species, which we believe is a good initial representation for this manuscript. That being said, we agree, that future studies will still need to continue to test species-specific host preference across additional behavioral trials. In much the same way, while we feel that the courtship assays we've included are typical measures of mating behavior, where we follow well-established protocols and include data for over 50% of all examined species, we also concur that more work is needed in the future to continue to verify and confirm our theories.

While our laboratory has some follow-up projects already in motion, we also strongly believe that our current manuscript provides a very detailed and robust first step towards supporting the existence of a tradeoff, and towards examining the potential behavioral ramifications of sensory bias across this genus of insects.

For example, a central premise of the work is that the size of a sensory structure should linearly correlate with the behavioral relevance of that sensory modality for any particular species. The authors suggest that the density of neurons in these structures is relatively constant across species, implying that the increased size is accompanied by an increase in the number of sensory neurons. This fails to address the key point—quantitative differences in sensory neuron number do not necessarily correspond to qualitative differences in odor discrimination or sensitivity. One could argue that the diversity of olfactory receptors and not the number of sensory neurons is more relevant to a high functioning olfactory system. While the authors

highlight the ecological framework of their work, their behavioral assays are too coarse to reveal any insight into how the selective pressures that may drive sensory specialization in different species that inhabit distinct niches.

We thank the reviewer for bringing this up! With our current data, while we cannot refute your argument that diversity of receptors could theoretically be as or more important than the sheer number of neurons, we can at least point out that most of the described *Drosophila* species have already been shown to have roughly identical types and diversity of receptors, both in regard to visual and olfactory signal reception. For example, all documented *Drosophila* species have the same 5 ommatidium types (Posnien et al. *PLoS One*, 2012; Hilbrant et al. *BMC Evolutionary Biology* 2014), though variation exists in the ratio of expression of each of these ommatidia types between species. As another example, although *D.suzukii* has 2.5x larger eyes than *D.melanogaster*, it is unclear whether this increase in the number of ommatidia is uniform, or if this increase in ommatidia represents an expansion of a particular rhodopsin type (e.g. perhaps contrast or color vision). It has also been shown that the number of ommatidia and their diameter are major determinants of the visual sensitivity and acuity of the compound eye of *Drosophila* (Gonzalez-Bellido et al. *PNAS* 2011), and for other insects such as Lepidopterans (Stockl et al., *Scientific Reports*, 2016; DOI: 10.1038/srep26041), where larger eyes correlate with enhanced visual behaviors. Moreover, that the relative size of a sensory system often dictates its relevance to the ecology of the animal.

In much the same way, olfactory receptor types and the diversity of chemosensory receptors is largely the same across all documented *Drosophila* species. For example, research across the 12 most studied genomes from the *Drosophila* genus suggests roughly identical total numbers of olfactory (OR), gustatory (GR), and ionotropic (IR) receptors as well as the same number of olfactory binding proteins (OBPs) in each of these species (*Drosophila* 12 Genomes Consortium, *Nature*, 2007; Sanchez-Gracia et al. *Encyclopedia of the Life Sciences* 2011, <https://doi.org/10.1002/9780470015902.a0022848>).

Thus the diversity of receptors within the visual and olfactory system is more or less the same for all documented species within this genus of flies, and therefore we feel it is unlikely that the variation we see between visual and olfactory biased species could be explained by receptor diversity, though again, the ratio of expression for these receptors between species (whether visual or olfactory) has not been fully addressed and may prove to be important to examine in the future. For example, the drastic increase in the ab3 sensillum type (containing Or22a & Or85b) has been linked to specialization in the island species, *Drosophila sechellia*, which is able to utilize a normally toxic host plant.

Added to manuscript (lines: 381-389, 485-501)

Another concern is the lack of clarity of whether the tradeoff between these structures reflects a true developmental constraint. The authors cite a recent paper by Hassan and colleagues in bioRxiv to support their work, suggesting that they have not delved further into any developmental mechanism, “as our manuscript provides far more ecological, evolutionary, behavioral and morphological (internal and external) evidence for this sensory tradeoff”. This bioRxiv preprint does indeed support a tradeoff in one pair of species—*melanogaster* and *pseudoobscura*—through a nice mechanistic analysis of a single transcription factor. However, remarkably these two species show a very similar EF ratio in the current study (Fig 4A), raising concerns about the sensitivity of their analysis and relevance to this current work.

We continue to be excited that the recent bioRxiv preprint by Hassan's group is in support of the idea that a tradeoff exists between these two sensory structures (essentially corroborating our findings from a separate laboratory and from a more developmental genetics direction); however, we concur that the comparison of their data to ours is not ideal. While you mention in Figure (4A,C) that the EF ratio between *D.melanogaster* and *D.pseudoobscura* is quite similar (7.46 and 7.74, respectively), it is also quite important to point out that this data is a representation of the ratio of eye size divided by antennal size. Thus if you look at our raw measurements of these two species, then you see that *D. pseudoobscura* has about 30-35% larger eye size, which was the main trait Ramaekers et al (bioRxiv 2018) measured, and that their manuscript and ours are in agreement with the counts of ommatidia. It is also important to mention that our data still shows *D. pseudoobscura* as more visual than *D. melanogaster*, which echoes the work from Hassan's laboratory, but that these two species are also sandwiched within 60 other species in our current study, while their study only examines two species.

In addition, we do not feel *D. pseudoobscura* is a good direct comparison to *D. melanogaster*, given the poor phylogenetic connection between these more distantly related species (17-30 million years apart), and that other pairings would perhaps better tackle the genetic, ecological, and evolutionary pressures that underpin this sensory tradeoff (i.e. that *D. subobscura* or *D. affinis* would be a better comparison for *D. pseudoobscura*, while *D. simulans* or *D. sechellia* would be a better comparison for *D. melanogaster*). That all being said, again, we feel this bioRxiv preprint does provide ample molecular genetic evidence that is consistent with our hypothesis that a tradeoff exists between these two sensory structures across the entire genus, which we feel is one of the central premises of our paper (i.e. that a tradeoff is the most consistent explanation for our data). Here though we continue to concede that more work will be needed in the future to conclusively demonstrate that this same gene (or genes) dictates the observed inverse variation in sensory systems across the other 60 *Drosophila* species that we test in our present manuscript.

Additional discussion added to main text (lines: 412-417)

Thus, in conclusion, we continue to believe that our manuscript (A) provides a strong foundation and a first step towards addressing these fundamental questions concerning the evolutionary pressures that shape sensory systems through the generation of robust metrics of both vision and olfaction regarding 60+ species, and (B) provides the groundwork and first large-scale documentation of the potential existence of a sensory tradeoff between vision and olfaction. However, again, we concur and acknowledge that ultimately, additional research in the future will need to act as the crucible to test the validity of this tradeoff hypothesis across the entire genus of fly, especially as it pertains to development constraint (e.g. molecular genetics from more than just two species), or perhaps across other holometabolous insects. Moreover, we look forward to sharing our raw data and ideas with the scientific community in the hopes of creating an ongoing dialogue to drive forward the quest to answer some of these fundamental questions in ecology, evolution and developmental biology, such as the pressures and constraints that have shaped the nervous system.

Reviewer #3 (Remarks to the Author):

The authors have put in a lot of effort to address the concerns of the previous draft and have improved the manuscript, however a few concerns remain.

Although the authors have showed different lines of evidence to suggest a trade-off between the sensory organs, the strongest line of evidence still comes from the species correlations. While I think it is fine to

propose a trade-off the authors should acknowledge that they can not confirm a trade-off with their current data and while the Ramaekers et al lends support to this theory this has only been shown in 2 species.

Moving forward with the current manuscript draft, we have toned down the “conclusiveness” of our results in regard to this tradeoff, though we continue to highlight that our various results are all consistent with and in support of this tradeoff hypothesis, including those documented from the peripheral nervous system, the primary processing centers within the brain, as well as those results collected during our developmental assessment. We have also now included additional emphasis in the discussion section that future work is still required to provide the molecular genetic confirmation of this tradeoff theory across more species.

Additional discussion added to main text (line: 466, 485-501)

I don't think the authors adequately address reviewers 4's concern with the use of ratios. At the very least they could show whether or not there is isometry in the traits of interest. And while I appreciate the authors have taken great means to address many of the statistical concerns of the past draft, a multiple regression approach should not be that difficult, particularly considering the authors are already performing phylogenetic regressions using the caper program.

In accordance with the requests from reviewer #3 and reviewer #4, we have now provided an additional statistical assessment (including a multiple regression) of the validity of the usage of ratios for comparisons made between visual and olfaction sensory systems. First, we found that the eye and funiculus surface area measurements scale isometrically with respect to the measurements taken from the body and the head. Thus we feel it continues to makes sense to use the EF ratio as our primary trait given that there is no real allometry in our data. Moreover, we show that neither body size ($p = 0.2935$) nor head size ($p = 0.5901$) significantly correlate with this EF ratio trait (Supplemental figure 1 H). In addition, we have plotted the analyses of the residuals (Supplemental figure 1 H), as well as shared again the R code we utilized to provide these statistical measures of allometry. Lastly, we have also conducted a multiple regression analysis (using the EF ratio, eye, funiculus, body, and head measurements from all 62 species), and indeed again, the EF ratio does not correlate with body or head size in this multiple regression ($p = 0.354$ and $p = 0.295$, respectively). Overall we continue to feel that we can safely maintain the usage of our EF ratio, as this trait does not simply scale allometrically with body or head size. We would like to again thank the reviewers for suggesting this additional statistical evaluation of the usage of ratios, as we feel these new tests have strengthened and further support our interpretation of the data.

Added to main text (line: 146-149)

Added to methods (line: 935-950)

We now provide a curated R script (as supplement through the online library) that we used to test allometry and perform a multiple regression, which might help readers utilize our data for their own purposes.

Highlighted below is the summary of the linear model (taken from the R script included with the raw data):

```
# multiple regression
```

```
fit <- lm(dat$EF.ratio ~ dat$eye + dat$funiculus + dat$body + dat$head)
```

```
summary(fit)
```

```
Coefficients:
```

```
      Estimate Std. Error t value Pr(>|t|)
(Intercept)  8.670e+00  5.831e-01  14.869 <2e-16 ***
```

```
dat$eye      1.585e-05 1.028e-06 15.423 <2e-16 ***
dat$funiculus -1.501e-04 7.823e-06 -19.184 <2e-16 ***
dat$body     -2.421e-04 2.590e-04 -0.935 0.354
dat$head     1.675e-03 1.585e-03 1.056 0.295
```

Line 143 Did the authors account for multiple comparisons when performing correlations amongst traits

We did not account for multiple regressions in our initial assessment, but through the last two revision processes, additional statistical comparisons have been performed and are now included. We continue to provide all R code scripts in conjunction with the raw data, both to provide maximal transparency, and to allow additional or novel testing of the dataset in the future (for example, if new ideas become available).

Figure 1b What are the colour codes on the phylogeny.

Our newly generated molecular phylogeny used an initial color scheme that sought to delineate species subgroups using a similar color pallet to the frontal head images of the *Drosophila* species; however, we have now reduced the color emphasis for clarity within the phylogeny, though we still highlight some subgroups for separation and ease of reading through the species lists (using grey, monotone boxes).

Figure 2 F and Figure 5 F How were these species chosen?

We used stratified random sampling. Species were selected from the 62 total in order to represent as many subgroups within the phylogeny as was possible to work with given a feasible time frame.

Figure 2B Why are some species blue and some red? There is not enough care in the preparation of the figure legends to clearly state the detail and purpose behind the detail in the figures. If the authors do not have enough space I suggest they simplify their figures to avoid confusion.

We have tried to maintain the same color codes throughout the entire paper. Where blue colors indicate potential olfactory bias (larger antenna), while red is indicative of more visually biased species or sensory systems (larger eyes). We have added additional text to all the main figures and supplemental figures and/or their legends to try to increase the clarity of this color information. Thank you for pointing this out, as there is quite a bit of information posted in each figure, and we hope the newest revision is clearer in regard to color codes for the data provided.

Line 151 it is unclear to me and from the methods what unique measurements were made on the 6 species. I can gather from Figure 2 that these were the basiconic, coeloconic and trichoid, but then I am not sure why the authors present Figure 2B? they have sensillum and ommatidium counts for all 62 species?

These 6 species had all sensilla manually counted and divided into the three main morphological types (e.g. basiconic, coeloconic, trichoid), whereas the majority of other species only had trichoid sensilla counted, which were more easily accessible. The 6 focal species also had ommatidia numbers manually counted, whereas the visual systems of the other species were assessed with surface area measurements of the compound eye (which turned out was a good approximation for ommatidia number). Information on this has been added to the manuscript.

Additional information added to main text (line: 153-181)

Line 157 Which 6 species did they choose? Clearly not the ones in brackets because with mel and suzukii that would be 8 species? If the species they choose did not include the ones in brackets I suggest they remove these to avoid confusion.

We selected 6 species to focus on:

D.melanogaster, *D.suzukii*, *D.busckii*, *D.americana*, *D.funebris*, and *D.pseudotalamancana*

Initially we selected and compared species in pairs with similar sensory sizes (e.g. similar eye size or similar antenna size), including comparisons utilizing *D.funebris* and *D.americana* twice. However, during the revision process, as per the reviewer comments, we adjusted our statistics instead to compare all species using an ANOVA, rather than arranged in paired t-tests. To ease the reading of this section, we have adjusted the writing to more clearly state this new, more global analysis across and between all 6 species.

Line 213 Your phylogeny comprised 59 of the 62 species?

Quoted from the previous round of revision (within the response to reviewer comments):

“While we could only locate and assess nuclear and mitochondrial genes for 59 of the 62 species (e.g. we are missing 2 subspecies of *D.mojavensis* as well as another individual species, *D.montium*), we feel this reanalysis is adequate to extrapolate towards the complete list of 62 species where we have additional morphometrics elsewhere in the manuscript. Again, we do not find a phylogenetic correlation with our EF ratio, suggesting that the phylogeny does not explain the trait variation we observe.”

In order to generate the statistical measurements and phylogenetic correction that were requested in the initial revision, we first needed to generate a complete molecular phylogeny of our species, which we prepared prior to the first resubmission. To reiterate the above quotation, we could not get sufficient genetic material for *D.montium* (which died in our stock, and was no longer available to reorder from commercial sources). We also did not successfully get genetic material from 2 of the subspecies of *mojavensis* (*D.mojavensis baja* and *D.mojavensis sonorensis*). Thus we could only generate a molecular phylogeny for 59 of the total 62 species, and thus could only include 59 species in the phylogenetic corrections for the statistical assessments; however, we still provide all the raw data we gathered from all 62 species, for example: the eye, funiculus, body and head measurements, as well as EF ratios.

We have added more explanations of this to the figure 1 legend (line: 727-734)

Line 238 The wording around this is awkward, accounting for phylogeny did not alter the strength of the relationship between the two traits. This brings me to another point P values are relatively meaningless I am more interested in how much variation can be explained by the relationship, authors should report r^2 and slopes throughout the ms. Note phylogenetic controlled analyses should be the gold standard and reporting the r^2 , slope and P value from these analyses supersedes any standard regression analysis and is therefore all that needs to be presented. In general the authors can simplify this whole section to just say we accounted for phylogeny, there were no strong relationships and here was the result.

Thank you! We adjusted the text in this section in an attempt to streamline the data and statistics presented, where we focused primarily on the phylogenetic controlled analyses. We apologize, as we are new to this type of reporting, but we did not perform linear regressions on the wing pigmentation nor the courtship data. Here instead we used paired t-tests and an ANOVA (please refer to Supplemental Figure 3 H,I) to test these traits with phylogenetic correction. Thus we compare males and females of a species with and without wing pigment to EF ratio (Supplemental Figure 3 H) and we compare the three courtship types again to EF ratio (Supplemental Figure 3 I).

Please see lines: 239-248

However, we have added slope information in addition to the already available R^2 and p-values for the linear regressions used throughout the manuscript (in figures). Thank you again for providing this suggestion, and we hope this is now suitably corrected.

Line 297 How did the authors identify this multitude? Randomly I hope.

In Figure 5E and Figure 2F, we used stratified random sampling to select representative members from as many of the major phylogenetic groups within the genus as possible. The time investment for dissection, staining, labeling, imaging and measuring the eye-antennal disc meant that it was not feasible to generate data from all 62 species, thus in these experiments we did as many species as we could in the allocated timeframe for this manuscript. We find these developmental analyses of the eye-antennal imaginal disc to be very interesting, and we hope others will undertake a broader sampling of species within this genus (or a sampling of other insects, such as Lepidopterans) in the future.

Line 363 How well are the olfactory genes and pseudogenes defined in the 14 *Drosophila* species? And are the authors tests robust enough to say pseudogenes are not playing a role? I think the authors need to be careful in their conclusions here.

For this analysis (Supplemental Figure 1 J) we did not identify “pseudogenes” ourselves, but utilized previously published data from as many species as were available (*Drosophila* 12 Genomes Consortium, *Nature*, 2007; Sanchez-Gracia et al. *Encyclopedia of the Life Sciences* 2011; Ramasamy et al. *Genome Biology and Evolution* 2016). We then compared these results using similar methods brought forward in an earlier paper that argued olfactory pseudogenes were correlated with an observed tradeoff in the evolution of primate color vision (Gilad et al. *PLOS Biology* 2004); however, we did not see the same evidence in the *Drosophila* data as was suggested by this “pseudogene hypothesis”, though perhaps as more *Drosophila* species become accessible, future studies could continue to retest this idea utilizing a more robust dataset. But, in accordance with your suggestions, we have softened our conclusions in our manuscript, both in the results and discussion sections.

Line 401 This sentence is oddly worded and thus confusing. I think the authors need to also be careful not to make to strong conclusions about their data. How robust is the courtship data across the 62 species? I think there are interesting patterns but the authors can't explain everything with their data, it is better to own the limitations than make bold and potentially incorrect statements.

We agree with your assessment, and again, have reduced the strength of the wording of our conclusions, similar to the editor's suggested phrasing (i.e. to concluding that a tradeoff is consistent with the observed patterns), as well as tried to highlight the perceived limitations of our data and interpretations while providing avenues and/or ideas for future research directions. The courtship data represents 32 of the 62 species, or roughly 50% of all those examined, where again, we mention that additional work is still needed.

Discussion section (lines: 233-255, 485-501)

Line 461 To bold I don't think you can confirm this. You have some good evidence in support of this theory.

Adjustments have now been made to the writing in the concluding paragraph (see above). Thank you for your time and insights regarding this revision process. We continue to find our manuscript improved by your comments and suggestions, thus again, thank you for your time and energy, and we hope we have been able to properly address each of your concerns with the newest resubmission!

Reviewer #4 (Remarks to the Author):

I have read the revised manuscript and authors' response to my comments. I still really like the paper, it has got a lot of interesting work in it. I don't agree with everything the authors say/do, but that doesn't mean they are wrong or that the paper isn't ready for publication. For the benefit of the editor I summarise the response to my main comments and add any remaining thoughts.

I made three major comments in my previous review, below I summarise the actions taken by the authors for each of these:

1) a lack of phylogenetic correction: the authors have produced a new phylogeny of the species in their dataset and used two methods to estimate the phylogenetic signal of their traits of interest. Both suggest phylogenetic signal is low for the key sensory traits, which is reasonable justification for not correcting for phylogeny in these cases. In other analyses the authors also present results from phylogenetic and non-phylogenetic analyses. They've done a good job on this critical issue.

Thank you again for your efforts and suggestions which have guided us along during this process!

2) use of ratios: the authors do not refute my suggestion that ratios can be misleading but keep this measure in their main analyses. As far as I can tell they do not add any analyses to explore this issue further. My view remains that this is a not ideal, but it is common practice.

Copied from above comment to reviewer #3:

"In accordance with the requests from review #3 and reviewer #4, we have now provided an additional statistical assessment of the validity of the usage of ratios for comparisons made between visual and olfaction sensory systems. First, we found that the eye and funiculus surface area measurements scale isometrically with respect to the measurements taken from the body and the head. Thus we feel it continues to make sense to use the EF ratio as our primary trait given that there is no real allometry in our data. Moreover, we show that neither body size ($p = 0.2935$) nor head size ($p = 0.5901$) significantly correlate with the EF ratio (Supplemental figure 1 H). In addition, we have plotted the analyses of the residuals (Supplemental figure 1 H), as well as shared again the R code we utilized to provide these statistical measures of allometry. Lastly, we have also conducted a multiple regression analysis (using the EF ratio, eye, funiculus, body and head measurements from all 62 species), and indeed again, the EF ratio does not correlate with body or head size in this multiple regression ($p = 0.354$ and $p = 0.295$, respectively). Thus overall we continue to feel that we can safely maintain the usage of our EF ratio, as this trait does not simply scale allometrically with body or head size."

3) inference of trade-offs: the authors provided an interesting response to my suggestion that the evidence for developmental trade-offs is not particularly strong, and that ecology/selection could produce similar patterns. But I'm not entirely convinced by the response; it's true that the ecological data is insufficient to support this idea but I would suggest that the developmental data is as well. I don't agree the authors 'conclusively demonstrate' a developmental trade-off explains the interspecific data – they demonstrate a potential mechanism for such a trade-off which is highly intriguing, but they don't show that it restricts the action of selection. I've only skim read Ramaekers et al, but at first glance I'd probably make the same comment on that paper. I appreciate this is quite an adaptationalist argument but, to me, it seems reasonable to me to give both hypotheses (ecology/development) equal weighting and await new data in the

future.

In light of your recommendations, we have reworded much of the discussion section in order to reduce the strength of the conclusions we draw from our data analyses, at least in regard to ecological versus developmental causality (which we concur, is still open for debate), and we provide additional ideas for future research that might continue to determine the legitimacy of both the existence of a tradeoff and the mechanism(s) by which such an event might occur repeatedly across this phylogeny or in other insects.

Thank you again for your open dialogue regarding this inference of a tradeoff. While we concur that our manuscript does not demonstrate conclusively that the observed inverse resource allocation is in fact a tradeoff, specifically one that is necessitated by the sharing of a common developmental structure, we do feel that the observed result and proposed mechanism is quite well reinforced by our dataset. Moreover, alternative ecological explanations do not seem to be as well supported. To say that in another way, if you compare our observed differences in sensory structures to any ecological rationale, such as habitat usage, food/host preference or geographical isolation, then we still do not find any alternative pattern or explanation for the data that we have acquired which outweighs our proposed developmental constraint or tradeoff. For example, we see drastic differences in sensory structures across species that utilize similar landscapes such as those species found within islands, within mountain ranges, within tropical forests or within deserts. Thus the physical and ambient/abiotic elements of defined landscapes do not seem to correlate with sensory system bias. We see dramatic differences between species that overlap geographically, and differences between those that share common ancestry within our phylogeny. More specifically, we even observe pronounced differences in these two sensory structures within subspecies living in the same micro habitats and utilizing strikingly similar hosts (i.e. the four *D. mojavensis* subspecies, a model group for incipient speciation, which are all localized to the South Western United States and Mexico and are all cactophilic breeders) (please see, Richmond et al. *Biological Journal of the Linnean Society* 2013, for more details about this subspecies example, such as geographical and host overlap).

Thus while we cannot explicitly rule out the potential impact of ecological dynamics, such as habitat, host choice, or geography (mostly due to incomplete natural history for a variety of understudied species), we feel that these factors do not appear to match the dataset as well as development constraint in regard to explaining EF ratio variances across our 62 species. However, again, we agree that more studies of natural history need to be generated for a wider array of non-*melanogaster* species so that future research can more accurately assess potential ecological, developmental, and evolutionary pressures or constraints that shape the nervous system. We would also be very keen to see more studies of *Drosophila* behavior that combine several species simultaneously during testing, to continue to ascertain the possibility of interspecies competition as a driving force in the observed differences in sensory structures. As “chemical ecologists” we tend to see the world through olfactory lenses, but we feel it is important to continue to analyze additional sensory structures, especially as they pertain to host and mate decisions.

In regard to the preprint, Ramaekers et al. (bioRxiv, 2018), we are again very pleased that another laboratory has reached a similar conclusion about a sensory tradeoff, albeit using a different approach and across fewer species. In addition, we believe that the multitude of patterns and correlations we present in this manuscript (e.g. external morphology, neuroanatomy, courtship and host navigation behaviors, as well as development) are all in alignment with each other and are best explained by a sensory tradeoff, perhaps driven by competition between close relatives for a mate or host plant. Nevertheless, we have taken steps to soften the conclusions we draw from our data in the written manuscript in order to reflect the reviewers concerns.

4) The minor comments are well addressed and the data should now be freely available once published, so maybe I will find the time to play with it myself!

Thank you! We hope you find the time to look them over as well, and maybe find something new, for example, we did not do any measurements of the mushroom body or other brain regions...

In sum, comments 2 and 3 are left in a slightly unsatisfied state from my particular perspective. That said, there are mitigating circumstances as the data may not be easily available to follow these points up. Indeed, the authors seem to accept these limitations but rightly state that future work can build on their results/data to test the conclusions obtained from their data. As such, although I could continue to argue my case (ad nauseam...) the authors are entitled to state their own interpretation of the data and I don't think my disagreeing with it should impede publication of a generally very nice paper. I look forward to seeing it in print.

Minor comments

Line 156: in the 'pairs' are not monophyletic I wouldn't refer to them as pairs. I'd just say '6 species that include the range of variation seen across the genus' or something similar.

Agreed. We have modified our writing to be more in line with your assessment. Thank you.

Line 235: The phylogenetic behavioural tests are interesting, but I'm not sure what "light or dark settings" (line 234) is exactly, light level?

Correct, a wide array of literature has examined the effects of light level (lux intensity) on *Drosophila* species courtship, including in some cases, even the effects of different wavelengths of light. Here we utilized published data on as many species as we could locate, where courtship between the male and female was conducted in either illuminated arenas (e.g. visual), or in complete darkness (non-visual).

Results: I think the text for 'Phylogenetic correction of eye to funiculus ratio' is maybe in the wrong position, its currently on page 7 but I would consider integrating it with the first results section on page 4/5.

We rather would like to keep the order as it is for the following reason. Figure 4 includes three analyses with a phylogenetic correction based on the inverse correlations that we first introduced in Figure 2 and 3. Therefore, we would like to keep the current order of data presentation (and writing) to maintain the flow of the manuscript.

General: standardise p-values to 3 decimal places for neatness?

Agreed, good idea! We have now limited the decimal places for neatness and consistency.

Lines 243- 246: not sure this is phrased well, the data could have phylogenetic signal but still produce these results. I'd delete the sentence beginning "therefore, we..."

We have adjusted this section of the text and removed this sentence.

Lines 35-378: in primates there is good evidence this is not a 'trade-off' it's the result of independent selection pressures shaping olfaction and vision in different ecological niches, so I don't think this is

analogous to your interpretation of the data. I'd still suggest deleting this paragraph, I think it is weak and doesn't add anything to the paper.

We still believe it is important to document other animal systems in which sensory tradeoffs have been previously proposed and/or examined using similar olfactory & visual comparisons. However, we have now shortened this entire section and we hope it is more in line with your suggestions.

General: I'd use central brain throughout, and ditch 'hemisphere'. Its confusing to use two terms.

We concur, and the text has been adjusted. Thank you for this suggestion.

Fig 2F: is this regression Phylogenetically corrected? State in the figure legend

No. This regression statistic was not corrected based on phylogeny (and this fact has now been added to the figure legend). We hypothesize a phylogenetic signal for the EF ratio only later in the manuscript and test three different analyses using this trait (Figure 4). For the ease of the flow of the written manuscript, we would like to keep it as it is. We of course could add the analysis to the figure here, if the reviewers feel that it is necessary.

Fig 3: if these 6 species are not 3 monophyletic pairs (?) it is perhaps a bit misleading to present them like this?

In hindsight we wish we would have perhaps selected a different, monophyletic pair as another example in addition to the 6 species that we had selected; however, we still feel the ANOVA format that we have provided is suitable to more globally compare each of the species in Figure 3. In association with your comment, we have tried to remove the word "pair" from the manuscript, and instead focus on the comparisons among and between the various datasets for all 6 examined species. Would have really liked to have addressed a true "pair" like *D. pseudoobscura* and *D. subobscura* for example, especially given their prevalence in the literature over the last 12 months... hindsight is 20/20... maybe next time!!

Methods lines 953-962: I'm unsure what the p-values refer to, I think the authors did a test comparing an estimated phylogenetic signal to one where it was set to zero? If so, the Blomberg estimates still suggest there is some signal, so phylogeny should be accounted for.

As we mention in the text, it looks not particularly significant. Pagel's lambda is basically 0, and Blomberg's K value is far from being 1. On the cautious side, we have phrased it as "EF ratio is not strongly supported by the phylogeny" throughout the manuscript. Below is each mention in the text of these two statistical measurements:

Lines 769-770: "Two statistical tests (Blomberg K and Pagel lambda) reveal that this sensory trait is not strongly supported by the phylogeny ($K = 0.478$, $p = 0.041$; $\lambda = 7.102e-05$, $p = 1$)"

Here are also Blomberg's own words about his test when regarding K-values less than 1 (which is what we show in our data using his statistical measurement; $K = 0.4783$, $p = 0.041$):

(Blomberg et al., *Evolution*, 2003)

"A K less than one implies that relatives resemble each other less than expected under Brownian motion evolution along the candidate tree. This could be caused by departure from Brownian motion evolution, such as adaptive evolution that is uncorrelated with the phylogeny (i.e., homoplasy)."

Thus, in conclusion, we still feel confident that both Blomberg and Pagel assessments are in agreement that our primary trait, EF ratio, is not strongly tied to the phylogenetic relation of the *Drosophila* species.

However, again to be cautious, we ran tests both with and without phylogenetic correction for several of the other traits of interest (i.e. EF ratio, EF ratio vs. wing pigmentation, EF ratio vs. light/dark courtship); moreover, we have included those values and statistics during each trait assessment (e.g. with and without phylogenetic correction), and in each case, we did not find phylogenetic signal. Therefore in summary we have attempted to account for phylogeny throughout the manuscript, and repeatedly found it not to play a predictive role.

Sample sizes: unless I'm being stupid, I don't see the sample sizes in all the figures/figure legends.

Thank you for pointing out this concern. We have reviewed the figures and figure legends and tried to make sample size more obvious in the most recent resubmission materials.

Again, we would like to thank you for your continued efforts to enhance this manuscript by helping to clarify our results, observations and interpretations. We really appreciate all your hard work in regard to these multiple revision steps, and we feel your extensive input and directives have greatly strengthened our writing, analyses, and conclusions.

Reviewers' Comments:

Reviewer #3:

Remarks to the Author:

Most of the concerns of previous drafts have been dealt with adequately by the authors.

I have a couple of minor comments on the terminology and discussions around phylogenetic comparisons and methods is not quite right but also understand this isn't really their expertise and I am unsure whether I am just being picky. My suggestion to the authors is that they simplify their discussion of phylogenetic corrections (its not really a correction), given they find weak signal they should simply mention they considered phylogenetic associations as a driver of trait variation but they didn't find a relationship between phylogeny and trait variation. Nothing else really needs to be said
Line 217 – 226 I know to some degree I am being picky but the phylogeny can't account for something. A phylogeny is the relationship of relatedness between species. You found no relationship between traits and phylogenetic relatedness, suggesting that phylogenetic relationships were not driving the observed pattern in your trait. The rest is just confusing.

Line 225 Perhaps make this a new sentence as it isn't really related to the previous and make it clear why you did this comparison. I would also be very careful with using habitat or ecology here you tested only a very small part of the ecology of these species.

Line 242 A significant correlation after correction isn't a test for phylogenetic signal.

REVIEWERS' COMMENTS:

We thank the reviewer for his additional advice on our manuscript!

Reviewer #3 (Remarks to the Author):

Most of the concerns of previous drafts have been dealt with adequately by the authors.

I have a couple of minor comments on the terminology and discussions around phylogenetic comparisons and methods is not quite right but also understand this isn't really their expertise and I am unsure whether I am just being picky. My suggestion to the authors is that they simplify their discussion of phylogenetic corrections (its not really a correction), given they find weak signal they should simply mention they considered phylogenetic associations as a driver of trait variation but they didn't find a relationship between phylogeny and trait variation. Nothing else really needs to be said

We thank the reviewer for this advice and now use the suggested sentence in the result section.

Line 217 – 226 I know to some degree I am being picky but the phylogeny can't account for something. A phylogeny is the relationship of relatedness between species. You found no relationship between traits and phylogenetic relatedness, suggesting that phylogenetic relationships were not driving the observed pattern in your trait. The rest is just confusing.

We now use the sentence as suggested by the reviewer.

Line 225 Perhaps make this a new sentence as it isn't really related to the previous and make it clear why you did this comparison. I would also be very careful with using habitat or ecology here you tested only a very small part of the ecology of these species.

Text adjustments have been made according to your suggestions and now mention that additional studies on the ecology of the different species is needed.

Line 242 A significant correlation after correction isn't a test for phylogenetic signal.

We agree and have erased the terminology "test for phylogenetic signal".